# Low-degree mantle melting controls the deep seismicity and explosive volcanism of the Gakkel Ridge

Ivan Koulakov[1,2✉], Vera Schlindwein [3,4], Mingqi Liu [5], Taras Gerya [5], Andrey Jakovlev [1,6] & Aleksey Ivanov [2]

The world's strongest known spreading-related seismicity swarm occurred in 1999 in a segment of the Gakkel Ridge located at 85°E as a consequence of an effusive-explosive submarine volcanic eruption. The data of a seismic network deployed on ice floes were used to locate hundreds of local earthquakes down to ~25 km depth and to build a seismic tomography model under the volcanic area. Here we show the seismicity and the distribution of seismic velocities together with the 3D magmatic-thermomechanical numerical model, which demonstrate how a magma reservoir under the Gakkel Ridge may form, rise and trigger volcanic eruptions in the rift valley. The ultraslow spreading rates with low mantle potential temperatures appear to be a critical factor in the production of volatile-rich, low-degree mantle melts that are focused toward the magma reservoirs within narrow magmatic sections. The degassing of these melts is the main cause of the explosive submarine eruptions.

[1] Trofimuk Institute of Petroleum Geology and Geophysics SB RAS, Prospekt Koptyuga, 3, 630090 Novosibirsk, Russia. [2] Institute of the Earth's Crust SB RAS, Irkutsk, Russia. [3] Alfred Wegener Institute, Helmholtz Centre for Polar and Marine Research, Am Alten Hafen 26, D 27568 Bremerhaven, Germany. [4] Department of Geosciences, University of Bremen, Bremen, Germany. [5] Department of Earth Sciences, ETH Zurich, Sonneggstrasse 5, 8092 Zurich, Switzerland. [6] Novosibirsk State University, Pirogova 2, Novosibirsk, Russia. ✉email: KoulakovIY@ipgg.sbras.ru

Oceanic spreading centers are some of the key elements in the global plate tectonics system, where the new lithosphere is created and moves apart. Most spreading centers are located in deep-water oceanic areas, and their experimental surveys face serious logistical problems that are especially critical in the Arctic due to the perennial ice cover and harsh natural conditions. For these reasons, the divergent Gakkel Ridge in the Arctic Ocean, which exhibits many paradoxical phenomena, remains one of the most enigmatic zones on our planet.

The Gakkel Ridge in the Arctic Ocean is an ~1800 km-long spreading zone that separates the Eurasian and North American Plates (Fig. 1a). It is characterized by ultraslow divergence rates that range from 13 mm/y in the western part to 6.3 mm/y near the eastern end in the Laptev Sea[1,2]. The Gakkel Ridge is exceptionally deep compared to other spreading centers: the sea bottom along its axial valley reaches depths of more than 5000 m[3], whereas, in other spreading centers, these depths rarely reach 4000 m and are usually ~3000–3500 m[4].

Despite its slow divergence rates, the Gakkel Ridge exhibits high levels of seismic activity with large numbers of strong earthquakes with magnitudes of more than 3, which have been recorded for decades by global seismic stations[5]. In recent years, some progress in earthquake recording along the Gakkel Ridge has been achieved owing to the installation of several seismic stations on islands in the Arctic Ocean[6]. The deployment of seismic instruments on ice floes along some local segments of the Gakkel Ridge has made it possible to investigate the seismicity of the ridge at the local scale in more detail[7]. A particularly interesting conclusion from these studies is that some of the earthquakes have occurred at depths of up to 25 km. Similarly, deep seismicity is observed in a few other slow-spreading centers and is explained by the anomalously low temperatures present beneath the rift axis[8–10].

Also, in some locations, the Gakkel Ridge is characterized by high volcanic and geothermal activity[11,12]. In this study, we consider a local segment of the Gakkel Ridge at 85°E, where a submarine volcanic eruption occurred in 1999, which was outstanding in many respects. First, this eruption triggered the strongest earthquake swarm that has ever been recorded instrumentally in the mid-oceanic ridges in the world[13–15]. Second, traces of effusive and explosive eruption activity were detected on the sea bottom at depths of 3700–4000 m[16–18]. To produce explosions under such conditions, the magma should be strongly enriched with volatiles with contents of greater than 13%, which are very atypical for mid-ocean basalts[16]. Third, the seismicity here (as well as in other segments of the ultraslow Gakkel Ridge) was unexpectedly deep[18]. Robustly resolved seismicity was detected at depths of up to 30 km, which are far below the bottom of the crust[19].

During the AGAVE campaign, which was conducted in 2007 by the Swedish icebreaker Oden, a seismic network was deployed on three separate ice floes in the area of volcanic activity in the Gakkel Ridge at 85°E/85°N. The network data were used to identify several hundred local earthquakes and to build a preliminary tomography model based on only the P-wave arrival times[19].

Here, we revisit this unique seismological dataset to derive a new seismic tomography model based on both P- and S-waves, which has crucial importance for interpreting the derived anomalies in terms of tectonic and magmatic processes, as well as for accurately determining the locations of the seismicity in this area. Within this study, we also create a three-dimensional thermomechanical numerical model that simulates a realistic case of magma generation under ultraslow spreading conditions. Based on these two types of studies, we propose a scenario that explains the specific magmatic and seismic activities of the Gakkel Ridge.

## Results

In this study, we use data that were obtained from the local seismic network, which consisted of three arrays that were deployed on three different ice floes in the area of the Gakkel Ridge segment at 85°E[20]. Each array was composed of four seismometers that were installed together on a single ice floe with an array aperture of ~1 km. The arrays drifted over the rift valley, and when they left the area of interest, the stations were removed and redeployed on other ice floes (Fig. 1b). In this way, each array was redeployed twice. For most of the local earthquakes, we were able to identify the P-waves and frequently, the SP-waves (i.e., S-waves converted to P-waves on the sea bottom), which facilitated high-accuracy determinations of the earthquake locations. The presence of sufficiently strong converted SP-waves generated by local earthquakes was previously shown by numerical modeling[21]. An example of picking a local event, which demonstrates fairly clear SP wave arrivals, is presented in Fig. S1. In our work, we used the data from 117 sources with 1028 P and 498 SP wave arrival times, which were inverted by using a modified version of the local earthquake tomography code LOTOS[22]. More details regarding the data, algorithms, intermediate results of the tomography study, and verification are presented in the Method section.

The distributions of the P- and S-wave velocity anomalies are shown in the horizontal and vertical sections contained in the Supplementary material in Figs. S3, S4. Here, we mainly focus on the $Vp/Vs$ ratios (Fig. 2), as it was determined in many studies that this parameter is most sensitive to the fluid and melt contents[23]. The resolution of the model and source location accuracy was assessed by a series of synthetic tests (Figs. S5–S9). These tests have shown that the uncertainties related to the damping definition and the trade-off between the source and velocity parameter determinations do not allow us to provide exact numerical values for the seismic parameters; therefore, we cannot uniquely convert our model into petrological properties. Nevertheless, our resulting images look similar to those of a number of models that were constructed for a variety of onshore active volcanoes, where the magmas and geothermal sources were investigated independently by using different approaches[24–26]. Based on these resemblances, our tomographic model reveals some structures that are likely associated with the volcano plumbing system and tectonic processes that are present along the spreading center.

The striking feature of the $Vp$ and $Vs$ anomaly distributions is their nearly perfect inverse correlation, which is a typical relationship that is observed in many active magmatic systems[24,25]. Indeed, P-wave velocities are more sensitive to rock compositions and are normally higher in magmas arriving from the mantle than in crustal rocks[25]. On the other hand, the S-wave velocities are mostly affected by the presence of liquid phases (e.g., melts and fluids) and are usually lower in active magma sources that contain partially molten materials and dissolved fluids. These two factors result in high $dVp$, low $dVs$, and very high $Vp/Vs$ ratios in areas with active magma sources that are located beneath volcanoes[24–26].

At shallow depths, the most prominent feature is anomaly "1" with high $Vp/Vs$ ratios, which is located in the central part of the rift valley in an area with recent signs of volcanic activity (Fig. 2). In the vertical sections, this anomaly extends to a depth of ~5 km below the sea bottom. According to the existing results of seismic surveys obtained along with other segments of the Gakkel Ridge, the crustal thicknesses in the axial areas rarely exceed 3.5 km[27];

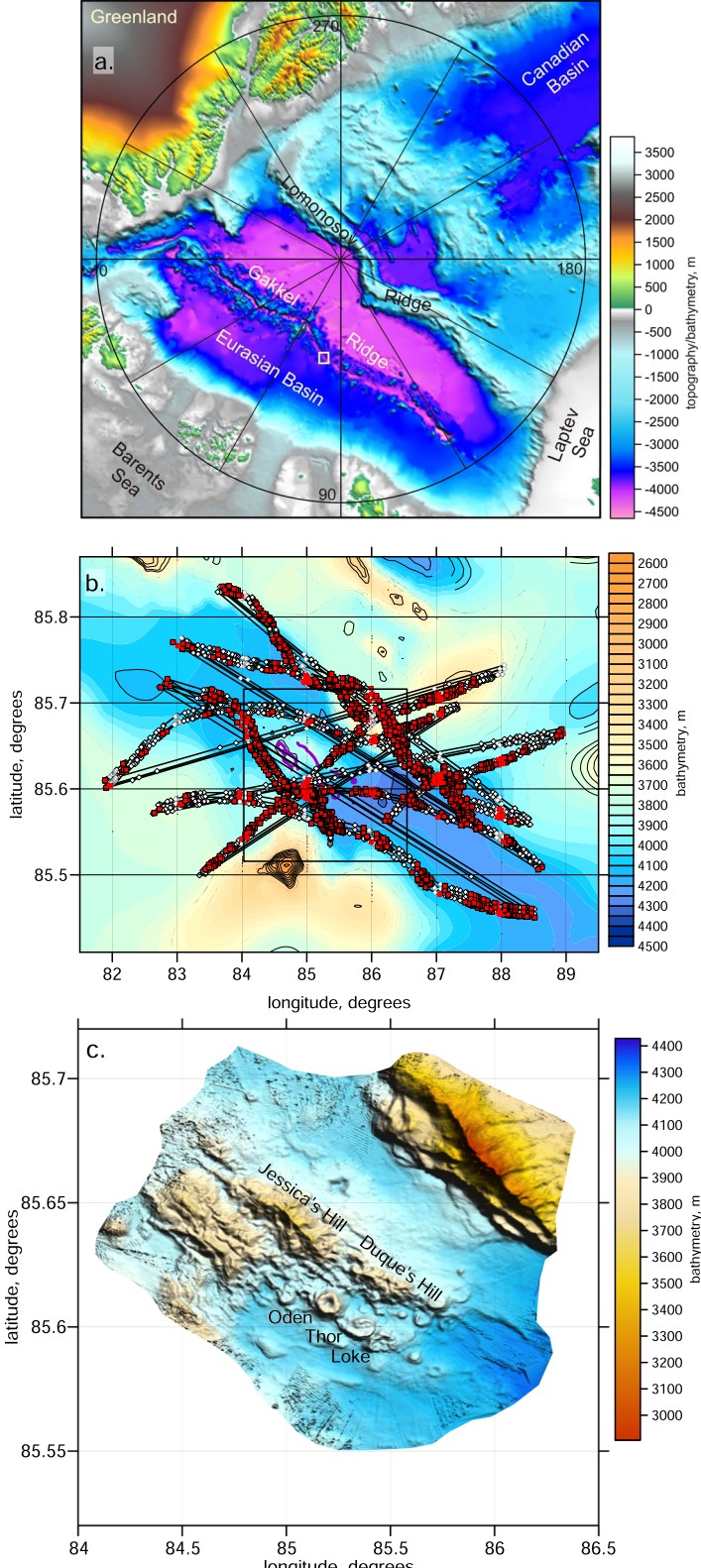

**Fig. 1 Study area. a** Bathymetry of the Arctic Ocean. The target area of this study is indicated by the white rectangle. **b** Tracks of the station migrations (thin lines). Red points indicate the station locations when they recorded local earthquakes. The background shows the bathymetry. **c** The detailed relief and major geological structures in the volcanic area at 85°E in the area are indicated by the rectangle in **b**. Bathymetry data from ref. [2].

thus, anomaly "1" appears to occupy the entire vertical crustal extent in the rift valley. Note that very few shallow events have occurred within the crust in this area. We propose that the high $Vp/Vs$ ratios within the rift valley at shallow depths are due to the

strong fracturing of rocks and seawater saturation. The high $Vp/Vs$ ratios at crustal depths might also be associated with remnant magma pockets that correspond to previous eruptions. Both of these factors cause the crustal rocks within the rift valley to be too

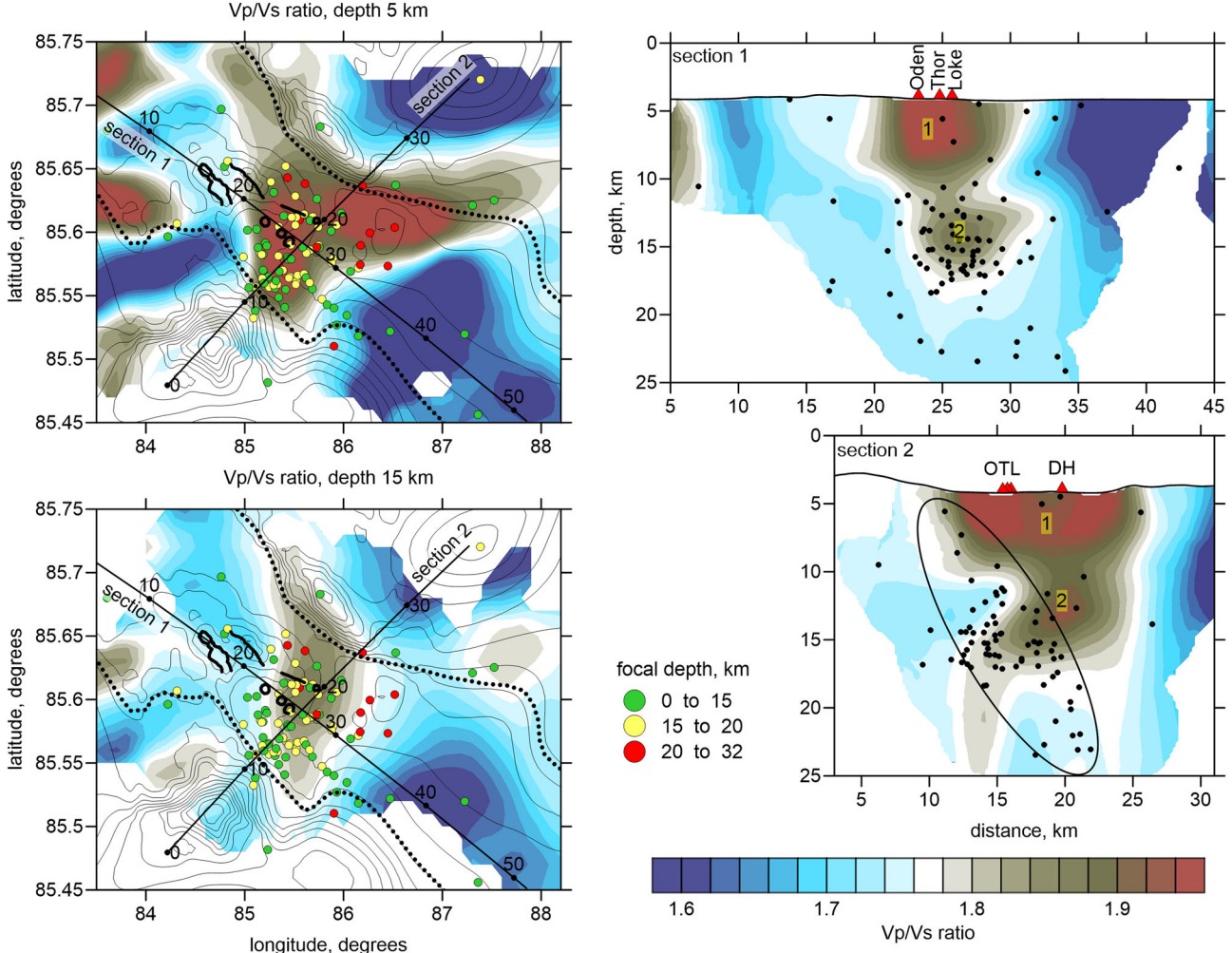

**Fig. 2 The *Vp/Vs* ratio distributions in two horizontal and two vertical sections.** The colored dots in the horizontal sections indicate the locations of earthquakes arranged by depth. The dotted lines indicate the border of the rift valley. The major volcanic structures are highlighted with solid black lines. The bathymetry is represented by thin contour lines (every 100 m). In the vertical sections, the black dots show the locations of events at distances less than 7 km. The red triangles depict the volcanic structures (e.g., OTL—Oden, Thor, Loke; DH—Duque's Hills).

soft, which prevents the accumulation of stresses and the generation of seismicity. In this case, the crustal extension along the rift axis occurs in a broadly distributed ductile manner.

The seismicity shown in the map view (left column in Fig. 2) is mainly located in the central part of the rift valley, where most volcanic manifestations are found, and corresponds to an area an anomaly consisting of high *Vp/Vs* ratios. In the vertical sections, most of the seismicity is confined within the anomaly at depths between 10 and 16 km b.s.l. Furthermore, the seismicity appears to extend downward to a depth of ~25 km. There are a few deeper events down to ~32 km, but their locations are less robust and should be considered with prudence. As highlighted by the ellipse in Section 2 in Fig. 2, the seismic events appear to be aligned along an inclined zone that dips from the southwestern border of the rift valley to the northeast. A possible interpretation of this distribution might be the existence of a fault that crosses the entire lithosphere. Similar features were found in the Dragon Horn area on the ultraslow Southwest Indian Ridge[28], where the seismicity down to ~15 km is associated with detachment faulting and deep high-temperature hydrothermal circulation.

The area with high a *Vp/Vs* ratio anomaly in the depth interval of 10–15 km b.s.l. is marked by "2" (both vertical sections in Fig. 2) and may represent a stable deeper reservoir that provides

the magma material for eruptions that occur within the rift valley. Section 2 shows that at a distance of 14 km along with the profile, we observe a downward elongation of this anomaly that might be interpreted as a conduit for delivering magma material from the asthenosphere. This magma reservoir may be considered as the main source of the eruptions occurring in the rift valley.

**Numerical magmatic-thermomechanical modeling of Gakkel Ridge volcanism.** To investigate the formation of the spatially localized magma plumbing system in the study area, 3D thermomechanical numerical modeling of an ultraslow spreading ridge was performed (see details in the Methods section). Our modeling results show that ultraslow spreading rates and low mantle potential temperatures are critical for the formation of a very low magma supply that is provided by volatile-rich, low-degree melts that were derived from the slowly rising and decompressing asthenosphere. In contrast to models that have the high magma supply characteristics of greater mantle potential temperatures and/or faster spreading rates (Figs. S17, S18), ultraslow spreading with a low magma supply is associated with strong variations in the brittle–ductile boundary depths along the ridge, which therefore spontaneously breaks into narrower and hotter magmatic sections (ca. 10% of the ridge length, Figs. 3a,

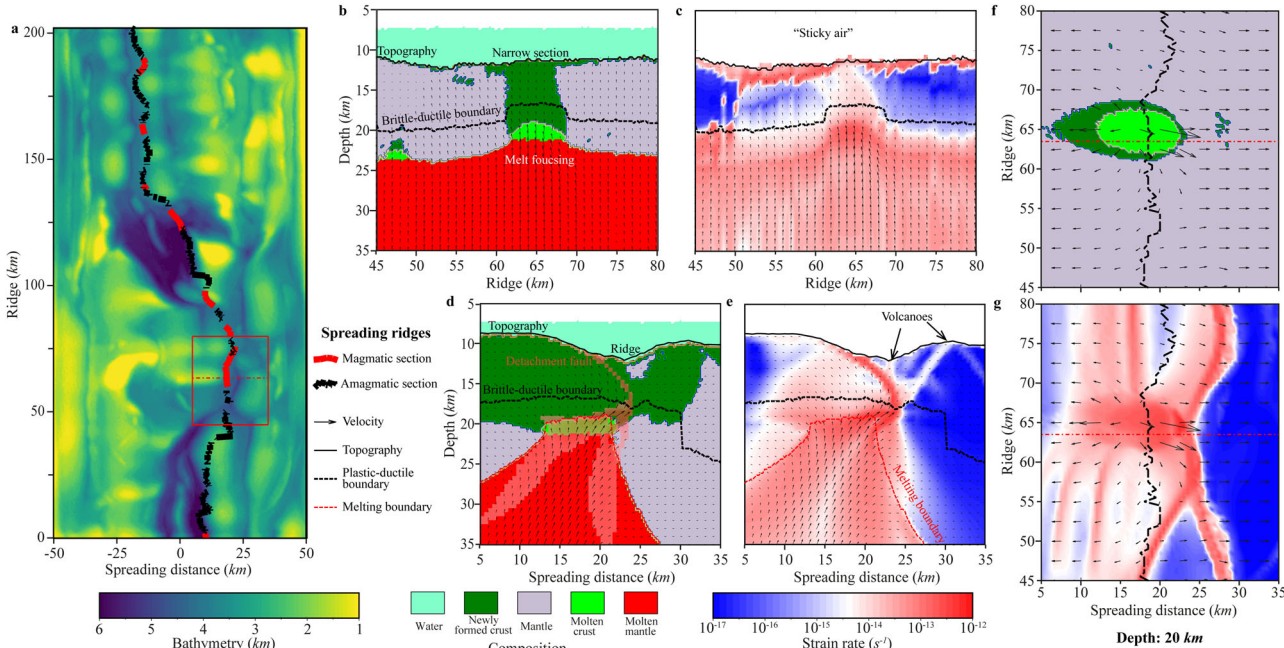

**Fig. 3 Numerical modeling of ultraslow spreading ridges at an evolution time of 10.4 Myr. a** Bathymetry. The red rectangle shows the area plotted in **b**–**g**. The dashed red line in the red square shows the profile location (ridge = 63.5 km) in **d** and **e**. The thick dashed red and black lines show the magmatic and amagmatic sections along the spreading ridge, respectively. **b**, **c** Profiles of the compositions and second invariant of the strain rate tensor along spreading ridges. **d**, **f**. Profiles of the compositions and second invariant of the strain rate tensor along the dashed red line contained in the rectangle. The high strain localization of the hanging wall in **e** may provide the channel for melt migration and consequently form volcanoes at the seafloor. **f**, **g** Horizontal slices showing the compositions and second invariant of the strain rate tensor at a depth of 20 km. The dashed black lines show the locations of spreading ridges. The very thick crust is likely due to the overfocusing of crustal growth that is introduced by the current simplified melt extraction and transport algorithm (see discussion in Methods).

S17, S18) and wider and colder amagmatic (ca. 90% of the ridge length, Figs. 3a, S17, S18) sections. Due to the ultraslow spreading rates and lower mantle potential temperatures compared to normal spreading ridges, the average magma supply along ultraslow spreading ridges is, on average, at least ten times less than those along ridges that are spreading at faster rates and/or with higher mantle potential temperatures. This implies a corresponding increase in the volatile contents (as volatiles are predominantly partitioned into low-degree melts, as is suggested by experimentally calibrated mantle melting models[29,30]). The deep, partially molten, rising asthenosphere materials are focused toward narrow, hot magmatic sections and induce high levels of volcanic and geothermal activity with a deep axial valley and shallow brittle–ductile boundary (Fig. 3a–c, Figs. S13, S14). In contrast, magmatic activity is rare in the colder sections with deep brittle–ductile boundary. Models that were created for a wide range of spreading rates and mantle potential temperatures show that with increasing spreading rates, even though the magma supply is very low due to the low mantle potential temperatures (e.g., 1255 °C), the amagmatic section still disappears at intermediate spreading rates (e.g., 40 mm/yr full rate, Fig. S17). Furthermore, with increased mantle potential temperatures, even though the spreading rates are ultraslow (e.g., 10 mm/yr full rate), owing to the large magma supply, the amagmatic section still disappears with high mantle potential temperatures (e.g., 1470 °C, Fig. S18). Our numerical experiments thus suggest that the Gakkel spreading ridge is developing under the combined effects of an ultraslow spreading rate and low mantle potential temperature and consists of several successive segments, which are connected by wide, oblique amagmatic sections (Fig. 3a).

The numerical modeling results presented in profile views along spreading direction (e.g., Fig. 3d, e, Figs. S7, S8) show that

magma with focused, rising, partially molten mantle is brought up to accrete on the footwall crust through detachment faults. A focused, hot, fast upwelling volatile-rich basaltic melt cuts the hanging wall plate, which results in volcanic eruptions in the rift valley. In a horizontal slice at a depth of 20 km (Fig. 3f, g and Fig. S11), an ovoid, partially molten, lower-crustal magma chamber appears, which likely represents strong fluid-melt activity and further indicates the main source of the volcanic eruptions that occur. Hence, we conclude that ultralow spreading ridges with low mantle potential temperatures may not only focus the deep, rising, partially molten asthenosphere toward the magma chamber to form narrow magmatic sections but also would subsequently trigger volcanic eruptions in the rift valley due to the very high volatile contents in the focused, low-degree, basaltic melts, as was previously proposed by Sohn et al. based on geochemical and petrological evidence[16]. The geodynamic modeling results also indicate that the lower portion of the detachment footwall near the bottom of the brittle deformation region located inside the crust is subjected to compressive stress and deforms internally as the fault rolls over to low angles before emerging at the seafloor (Fig. 3d, e). Then, the bending of the detachment faults triggers reverse faulting in an extensional detachment system, which is consistent with previous seismic observations obtained at 13°N on the Mid-Atlantic Ridge[31].

Although the thermomechanical model was constructed independently of the seismic tomography results, the configurations of the main structures appear to be similar for these two cases. In particular, the strain rate distribution shown in Fig. 3e demonstrates asymmetrical patterns, including an inclined zone of normal faults, which is consistent with the seismicity distribution (Section 2 in Fig. 2); the asthenospheric upwelling in the mantle is shifted with respect to the location of the rift valley, which would correspond to

the deep anomaly with high $Vp/Vs$ ratios. In the horizontal section at 15 km depth, the anomaly with high $Vp/Vs$ ratios and an elliptical shape appears to be consistent with the map view of the compositional features and strain rates in the numerical model (Fig. 3f, g). These similarities demonstrate that the physical processes simulated by our numerical model appear to be realistic and sufficiently generic to explain the outstanding magmatic and seismic activities in the examined segment of the Gakkel Ridge.

## Discussion

The numerical models demonstrate that ultraslow spreading causes relatively low average temperatures in the mantle beneath the ridge, which in turn is a reason for the low degree of melting. We estimate that in the case of the Gakkel Ridge, the maximum degree of mantle melting under the ridge is ~9–11% (cf. degree of melting inside the melt boundary in Fig. S16), whereas, in most other spreading zones with much faster extension rates, the degree of melting may reach 15–20%[32]. Because of such a low degree of melting, the magma beneath the Gakkel Ridge is focused in discrete zones that are separated by long amagmatic segments. This appears to be a common feature for most ultraslow and slow-spreading centers, in contrast to fast-spreading zones, where magma generation occurs continuously along the ridges[33]. In our numerical analysis, we see that the area of magma generation (partially molten lower-crustal magma chamber, Fig. 3b, d, f) has a lateral extent of ~7 km and elliptical shape in the map view, which are consistent with the size of the high $Vp/Vs$ anomaly observed in our tomography model (map view at a depth of 15 km in Fig. 2). In the vertical section of the numerical model along the ridge (Fig. 3b, c), we see that magma pockets accumulate in zones of the thinned lithosphere. This means that if there is any melting in the asthenosphere in other segments of the ridge, the produced melts percolate along the bottom of the lithosphere[29,30] (see Supporting Information) toward the discrete zones of magma accumulation. This creates a positive feedback mechanism in which the larger the amounts of molten material that accumulate in the pocket, the warmer and thinner will be the lithosphere in this zone, and the more active migration of melts along the lithosphere boundary will occur. In this case, a relatively small amount of molten material may absorb volatiles from a large volume of asthenosphere beneath the entire ridge and carry them to the zones of magma accumulation (Fig. 3b, d). As a result, due to the high melt/solid partitioning coefficients of the volatiles[29,30], the volatile contents in such sparse magma centers should be one order of magnitude higher than those in regular continued melt-rich spreading zones[29]. Thus, there is a paradoxical situation: low temperatures and low melting degrees lead to high concentrations of volatiles, which in turn lead to exceptionally strong, explosive eruptions in discrete volcanic centers on the ridge.

Interpretations of the results of tomographic inversion and numerical modeling should consider the existing geochemical data. A study of the olivine-hosted melt inclusions at the segment of the Gakkel Ridge at 85°E yielded measured $CO_2$ concentrations between 167 and 1596 ppm[34]. The $CO_2$ concentrations are not correlated with the concentrations of other chemical components, and this observation can be explained by $CO_2$ degassing from the melt inclusions. The $CO_2$ concentrations in the primary melt can also be estimated from the Ba concentrations measured in glasses (by considering them as representatives of melts) and assuming $CO_2/Ba = 81.3$ as the characteristic of global MORBs[35]. By using the Ba concentration data of[34] and $CO_2/Ba = 81.3$, the $CO_2$ concentrations are estimated to be $1514 \pm 84$ (1 SD). This value overlaps with the highest measured values in the melt inclusions. The saturation pressures for $CO_2$ concentrations of 1500–1600 ppm are 3.0–3.1 kbar[36]. Considering a water depth of 4 km and

crustal thickness of 5 km, such saturation pressures correspond to a depth of ~13 km below sea level, which in turn overlaps with the anomalously high $Vp/Vs$ ratios of anomaly 2 (Fig. 2). Thus, anomaly 2 likely marks a region of $CO_2$ degassing within a magma chamber. Small blobs of recycled material, which are the $CO_2$ source, are probably heterogeneously distributed in the asthenosphere. Such materials are the source of $CO_2$-rich silicate melts that are produced at low degrees of partial melting[37].

Based on the obtained models and available data, we propose a scenario for the tectono-magmatic activity on the Gakkel Ridge, which is illustrated schematically in Fig. 4. In this scheme, the background consists of the vertical section of the $Vp/Vs$ ratios across the rift valley (Section 2 in Fig. 2), which corresponds to the section of the numerical model shown in Fig. 3d, c. We propose that beneath the segment of the Gakkel Ridge at 85°E, there is asthenospheric upwelling, which is observed at a depth of 15 km as an elliptical body with high $Vp/Vs$ ratios. Although the melting degree in this area is low, it is anomalously contaminated with volatiles. At some critical volatile concentration, it begins to transform into a gaseous phase. Because of the relatively low temperatures, the mantle lithosphere around the asthenospheric pocket remains cold and behaves as a semibrittle material. Active degassing of volatiles from the rising decompressing melts leads to abrupt volume changes and creates fractures in the marginal areas of the lithosphere around the areas of ascending asthenosphere, which is marked by deep seismicity along an inclined zone, as shown in Fig. 4.

The released gas migrates upward to the stable magma reservoir located at a depth interval of 5–10 km below the sea bottom, which is revealed in our tomography model as a distinct anomaly with high $Vp/Vs$ ratios. The magma material in this reservoir is gradually saturated with volatiles, which corresponds to the quiet periods between episodic volcanic activity. Similar cycles that are caused by the episodic filling and emptying of upper crustal magma reservoirs are observed on some onshore volcanoes, such as Nevado del Ruiz[38] and Mount Spurr[28]. Many authors state that during the quiet periods of volatile accumulation, the magma reservoirs are covered by a rigid lid (e.g., the violet layer above the magma reservoir in Fig. 4), also called the self-sealed zone (SSZ)[39]. During the episodic volcanic activities, the high-pressure magma in the reservoir that is lightened by bubbles breaks the SSZ and escapes to the surface and thus producing explosive eruptions with the release of large amounts of gases.

In summary, in this study, with the use of local seismic observations and numerical thermomechanical modeling, we were able to provide a plausible scenario (Fig. 4) that explains the causes of the strong effusive and explosive volcanic eruptions and unusually deep seismic activity in the segment of the Gakkel Ridge at 85°E/85°N. We found a paradoxical feature that such exceptionally active volcanic and seismic behavior at the ultraslow spreading center is likely caused by a relatively cold asthenosphere with a very low degree of melting under the Gakkel ridge, which produces basaltic melts with anomalously high volatile contents. Degassing of the assembled low-degree melts inside the spontaneously formed, localized, rising asthenospheric region causes fracturing of the surrounding semibrittle cold mantle lithosphere, which triggers seismicity at depths of up to 25 km.

## Methods

**Seismic tomography.** For tomography, we used a version of the LOTOS code[22], which was specially adapted for a case of a floating network. Some modifications had to be done for the input format, which included the variable coordinates of the stations for each event. Both the P- and S-wave velocity models included the water layer with a constant velocity value equal to 1.44 km/s. The ray tracing was conducted by a 3D bending algorithm for the entire path between source and receiver including the water layer. For calculation of travel times, we used a lower resolution regional bathymetry model presented in Fig. 1c, which appeared to be adequate

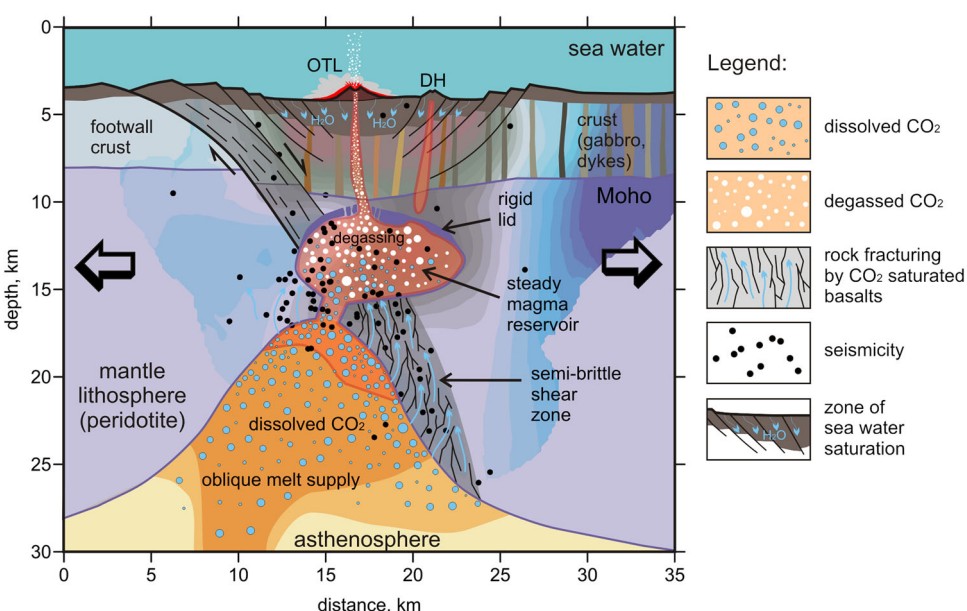

**Fig. 4 Interpretation of the resulting *Vp/Vs* ratios along a section that is oriented across the rift valley (vertical Section 2 in Fig. 2) by considering the numerical modeling results (Fig. 3d, e).** The black dots depict earthquakes. See details in the text.

regarding some natural averaging occurring due to a finite frequency content of the arriving waves.

The velocity distributions were parameterized by a set of nodes installed below the sea bottom taking into account the bathymetry information. The nodes were installed only in areas with sufficient ray coverage (more than 0.2 of average ray density). To minimize any effect of the grid geometry on the results, we performed the inversions in four grids with different basic orientations (0, 22, 45, and 66 degrees) and then averaged the results in a regularly spaced three-dimensional model. The inversion was performed simultaneously for the *dVp* and *dVs* anomalies, as well as for source relocation parameters (three coordinates and origin time corrections for each event), using the LSQR algorithm for solving large systems of linear equations[40,41]. The solutions were stabilized by two types of regularization: amplitude damping and smoothing. The corresponding controlling parameters were determined with the use of synthetic modeling. After computing the 3D velocity model, it was used to relocate the events using the bending ray tracing, and the procedure of inversion was repeated again. In total, we performed five iterations including relocation and inversion steps.

As a result of the iterative tomography inversions, we have obtained the 3D distributions of the *P*- and *S*-wave velocities, as well as coordinates of events, relocated in the derived 3D models. In the context of this study, we find it important to present the distributions of the *Vp/Vs* ratio, as it was found in many studies that this parameter is sensitive to the content of fluids and melts[23]. In our case, the *Vp/Vs* ratio was derived by the division of the resulting absolute velocities of the *P*- and *S*-waves. The adequacy of this method was checked by a series of synthetic tests.

Synthetic modeling is an important stage of the tomography workflow as it gives estimates for the spatial resolution of the recovered structures, as well as optimal values for the inversion controlling parameters. In our case, in synthetic modeling, we simulate the same workflow as used to process the experimental data. The synthetic velocity model is defined as a superposition of the 1D absolute reference velocity and 3D anomalies. The synthetic travel times are calculated for the same stations and source distributions as derived after obtaining the main tomography model. Then we "forget" the coordinates and origin times of the sources and perturb the arrival times with realistically distributed noise with the average deviations 0.03 s and 0.03 s for the *P*- and *S*-waves. The recovery procedure is identical to the case of experimental data inversion including the step of preliminary source locations using the grid-search method. While performing the recovery of the synthetic model, we adjust the main inversion parameters to obtain the best quality of reconstruction; these parameters are then used for the processing of the experimental data. The results of the recovery of several synthetic models are presented in Supplementary Materials.

**Magmatic-thermomechanical modeling.** A 3D self-consistent magmatic-thermo-mechanical numerical model based on code I3ELVIS is used, which combines finite difference on a fully staggered Eulerian grid and a marker-in-cell technique[42]. The momentum, mass, and heat conservation equations are solved on the non-deforming Eulerian grid whereas the advection of transport properties including viscosity, plastic strain, temperature etc. is performed with the moving Lagrangian markers[32].

Lagrangian markers leave the Eulerian model domain through the left and right lateral boundaries and are then recycled through the top and the bottom of the model as seawater and mantle markers, respectively. This Eulerian-Lagrangian numerical modeling scheme with open boundaries allows for an infinitely long plate separation with the use of a laterally limited Eulerian computational domain and a relatively small amount of continuously recycled Lagrangian markers.

The viscous and brittle/plastic properties (Table S2) are implemented via evaluation of the effective viscosity of the material. Brittle/plastic rheology is controlled by fracture-related strain weakening[32]. It is implemented by using a Drucker-Prager criterion:

$$\eta_{\text{plastic}} = \frac{\sigma_{\text{yield}}}{2\dot{\varepsilon}_{II}}, \tag{1}$$

$$\sigma_{\text{yield}} = C_\gamma + \varphi(P - P_f), \tag{2}$$

where $\eta_{plastic}$ is the effective viscosity of plastic rheology. $\sigma_{yield}$ is the yield stress (Pa). $\dot{\varepsilon}_{II}$ is the second invariant of strain rate tensor. $P$ is the dynamic pressure on solids (Pa), $P_f$ is the hydrostatic fluid pressure $\left(P_f = \rho_f gy, \rho_f = 1000\,\text{kg/m}^3\right)$, $C_\gamma$ is the plate strength at $P - P_f = 0$ (for both tensile and confined fracture) that depends on the plastic strain $\gamma$. $C_0$ and $C_1$ are the initial and final strength values for the fracture-related weakening, respectively. $\gamma_0$ and $\gamma_1$ are the upper and lower strain limits for fracture-related weakening, respectively. $\varphi$ is the internal friction coefficient. $\varphi_0$ and $\varphi_1$ are the upper and lower limits, respectively. Strain weakening is obtained by strain-dependent linear interpolation between the parameterized minimum and maximum values of $\varphi$ and $C_\gamma$:

$$\varphi = 1, \text{ when } P < P_f \text{ (tensile fracture))}, \tag{3a}$$

$$\left.\begin{array}{l} \varphi = \varphi_0, \text{ for } \gamma \leq \gamma_0 \\ \varphi = \varphi_0 + (\gamma - \gamma_0)\frac{\varphi_1 - \varphi_0}{\gamma_1 - \gamma_0}, \text{ for } \gamma_0 < \gamma \leq \gamma_1 \\ \varphi = \varphi_1, \text{ for } \gamma > \gamma_1 \end{array}\right\}, \text{ when } P \geq P_f \text{ (confined sfracture)}, \tag{3b}$$

$$C_\gamma = C_0, \text{when } \gamma \leq \gamma_0, \tag{4a}$$

$$C_\gamma = C_0 + (\gamma - \gamma_0)\frac{C_1 - C_0}{\gamma_1 - \gamma_0}, \text{when } \gamma_0 < \gamma \leq \gamma_1, \tag{4b}$$

$$C_\gamma = C_1, \text{when } \gamma > \gamma_1, \tag{4c}$$

Integrated plastic strain $\gamma$ is a scalar value time-integrating the second invariant of the plastic strain rate tensor $\dot{\varepsilon}_{ij(\text{plastic})}$:

$$\gamma = \int_{t'=0}^{t} \sqrt{\frac{1}{2}\left(\dot{\varepsilon}_{ij(\text{plastic})}(t')\right)^2}\,dt', \tag{5}$$

where $t$ is time, $t'$ represents each time before time $t$, $dt'$ is the time step just before each time $t'$. Strain weakening assumed in the model is similar to those in previous numerical studies of mid-ocean ridges. It is related, in particular, to water and melts percolation along

fault zones and their intense serpentinization that strongly decreases the strength of fractured fault rocks.

For the ductile rheology, the contributions from different flow laws such as dislocation and diffusion creep are considered by composite rheology for $\eta_{ductile}$:

$$\frac{1}{\eta_{\text{ductile}}} = \frac{1}{\eta_{df}} + \frac{1}{\eta_{ds}}, \tag{6}$$

$\eta_{df}$ and $\eta_{ds}$ are effective viscosities for diffusion and dislocation creep, respectively. For the crust, constant grainsize is assumed and $\eta_{df}$ and $\eta_{ds}$ are computed as:

$$\eta_{df} = \frac{A_D}{2\sigma_{cr}^{n-1}} \exp\left(\frac{E + PV}{RT}\right), \tag{7}$$

$$\eta_{ds} = \frac{1}{2} A_D^{\frac{1}{n}} \exp\left(\frac{E + PV}{nRT}\right) \dot{\varepsilon}_{II}^{\frac{1}{n}-1}, \tag{8}$$

Where $R$ is gas constant, $P$ is pressure, $T$ is the temperature (in K), $\sigma_{cr}$ is the assumed diffusion-dislocation transition stress, and $A_D$, $E$, $V$, and $n$ are experimentally determined flow-law parameters, corresponding to the material constant, the activation energy, the activation volume, and the stress exponent, respectively.

For the mantle, the ductile creep model also takes into account grainsize reduction and growth processes assisted by Zener pinning[43–45]. The rheology follows a composite law, wherein

$$\eta_{df} = \frac{1}{2A_{df}} \left(\frac{\pi r}{2}\right)^m \exp\left(\frac{E_{df} + PV_{df}}{RT}\right), \tag{9}$$

$$\eta_{ds} = \frac{1}{2A_{ds}^n} \dot{\varepsilon}_{II}^{\frac{1}{n}-1} \exp\left(\frac{E_{ds} + PV_{ds}}{nRT}\right), \tag{10}$$

Where $E_{df}$ and $E_{ds}$ activation energy and $V_{df}$ and $V_{ds}$ activation energy, are for diffusion and dislocation creep, respectively. A is a prefactor that differs depending on the deformation regime. $n$ is the dislocation creep exponent. $r$ is the grain interface curvature, which is the average grainsize in two phases[43,46]. $m$ is the diffusion creep grainsize exponent. The factor $\left(\frac{\pi}{2}\right)$ is used to compute from the interface curvature to the actual grainsize. The interplay between diffusion and dislocation creep is controlled by a grainsize evolution equation dependent on the mechanical work and temperature. The grainsize evolution model relies on several assumptions:

(1) Mantle peridotite is assumed to be composed of two well-mixed phases—olivine and pyroxene with a fixed volume fraction of 60% and 40%, respectively. These phases are considered to have the same density and rheology.
(2) In both phases, the relative motion is considered to be negligible and therefore their velocities are the same.
(3) It is assumed that the grainsize distribution is close to a self-similar log-normal distribution. Therefore, it always retains the same shape and its mean-variance and amplitude are fully characterized by unique grainsize.

The system is in a state known as pinned state limit[43,47], wherein the grainsize evolution is controlled by the pinning of phases by each other (i.e., Zener pining is dominant). In these conditions, the grainsize is controlled by the roughness $r$ of the interface between the two phases. A relation between the mean grainsize $h$ (sufficient to fully describe the system) and the roughness $r$ is given by $h = \frac{r}{\sqrt{h_g}}$, where $h_g \approx \frac{\pi}{2}$ for the phase volume fraction in our model[46]. The roughness evolution is described by the following equations [43,44,47,48]:

$$\frac{dr}{dt} = \frac{\eta G_I}{qr^{q-1}} - \frac{f_I r^2}{\gamma_I \eta} \Psi, \tag{11}$$

$$G_I = \frac{G_g}{G_{fac}} \frac{q}{p} r^{(q-p)}, \tag{12}$$

$$G_g = A_g \exp\left(\frac{E_g + PV_g}{RT}\right), \tag{13}$$

$$f_I = f_0 \exp\left(-2\left(\frac{T}{1000}\right)^{2.9}\right), \tag{14}$$

where $G_I$ is interface coarsening, $G_g$ is grain-growth rate, $G_{fac} = 100$ is grain-growth rate factor, $q = 4$ is roughness coarsening exponent, $p = 2$ is grainsize coarsening exponent, $\gamma_I$ is the surface tension, $A_g = 2 \times 10^{(4-6p)}$ is pre-exponential factor, $E_g = 3 \times 10^5$ is grain-growth activation energy, $V_g = V_{df}$ is a grain-growth activation volume, $f_I$ is the fraction of mechanical work $\Psi$ converted to interface damage resulting in grainsize reduction; $f_0 = 0.001$ is interface damage at 1000 K, $\eta = 3\varphi_{ol}\varphi_{px}$ is interface area density depending on the volume fractions of olivine ($\varphi_{ol} = 0.6$) and pyroxene ($\varphi_{Px} = 0.4$) in the mantle.

The final effective viscosity is defined as the minimum value between the plastic equivalent viscosity ($\eta_{plastic}$) and the ductile viscosity ($\eta_{ductile}$), which is further controlled by the cutoff values of $[10^{18}, 10^{24}]$ Pa·s:

$$\eta_{\text{eff}} = \min(\eta_{\text{plastic}}, \eta_{\text{ductile}}), \tag{15}$$

During post-processing of numerical modeling results, the dominant rheological regime is calculated based on the relationship between different kinds of deformation regimes.

Plastic deformation, when $\eta_{plastic} \geq \eta_{ductile}$;
Ductile deformation, when $\eta_{plastic} < \eta_{ductile}$:

$$\begin{cases} \frac{\eta_{df}}{\eta_{ds}} \leq 0.1, & \text{Diffusion creep dominates} \\ 0.1 < \frac{\eta_{df}}{\eta_{ds}} \leq 1, & \text{Transition } I - \text{diffusion creep dominates, but with some dislocation creep} \\ 1 < \frac{\eta_{df}}{\eta_{ds}} \leq 10, & \text{Transition } II - \text{dislocation creep dominates, but with some diffusion creep} \\ \frac{\eta_{df}}{\eta_{ds}} > 10, & \text{Dislocation creep dominates} \end{cases}$$

**Mid-ocean ridge processes.** In order to simulate plate breakup and oceanic spreading, four key processes are implemented in a simplified manner[1]: thermal accretion of the oceanic mantle lithosphere resulting in the plate thickness growth[2], partial melting of the asthenospheric mantle, melt extraction and percolation toward the ridge resulting in crustal growth[3], magmatic accretion of the new oceanic crust under the ridge, and[4] hydrothermal circulation at the axis, resulting in excess cooling of the crust[32,49–54].

(1) Thermal accretion of the oceanic mantle lithosphere. A temperature-dependent viscosity for the non-molten mantle is used[55]. Consequently, the cooling asthenospheric mantle becomes rheologically strong and accretes spontaneously to the bottom of the oceanic lithosphere.
(2) Partial melting of the asthenospheric mantle, melt extraction, and percolation toward the ridge[32]. In our model, mafic magma added to the crust is balanced by melt production and extraction from melting. However, melt percolation is not modeled directly and is considered to be nearly instantaneous[51,56]. The standard (that is, without melt extraction) volumetric degree of mantle melting, $M_0$, changes with pressure and temperature according to the parameterized batch melting model of Katz et al.[30]. Lagrangian markers track the amount of melt extracted during the evolution. The total amount of melt, $M$, for each marker considers the amount of previously extracted melt and is calculated as $M = M_0 - \sum_m M_{ext,m}$, where $\sum_m M_{ext,m}$ is the total melt fraction extracted during the previous $m$ extraction episodes[57]. In the beginning, all mantle rocks are assumed to be melt-depleted $\sum_m M_{ext,m} = M_0$. The rock is considered non-molten (refractory) when the extracted melt fraction is larger than the standard one (that is, when $\sum_m M_{ext,m} > M_0$). If $M > 0$ for a given marker, then melt fraction $M_{ext} = M$ is extracted and $\sum_m M_{ext,m}$ is updated. The extracted melt fraction $M_{ext}$ is assumed to propagate much faster than the plates deform[56]. Hence, the instantaneous transmission of extracted melt to the magma chamber is reasonable. Extracted mantle melts first travel vertically until they rich mantle solidus surface under the lithosphere and then migrate along this surface[51] toward regions with the highest local surface topography (i.e., towards local melt traps) where they form multiple magma regions. In ultraslow ridges, such regions are typically located in multiple places beneath the ridge axis where oval-shaped magma regions spontaneously form by focusing on the extracted low-degree melts (Fig. 3). In order to ensure melt volume conservation and account for mantle compaction and subsidence in response to the melt extraction, melt addition to the bottom of the magma region is performed at every time step by converting the shallowest markers of the hot partially molten mantle into magma markers. In each magma region, the volume of these new magma markers matches the local volume of extracted assembled melts computed for the time step. It should be noted that our simplified melt transport treatment produces very thick (up to 10 km) crust in magmatic segments of ultraslow spreading ridges (Fig. 3). This is likely due to the overfocusing of crustal growth by our simple melt extraction and transport algorithm, which brings all extracted melts to very localized magma chambers forming at the maxima of mantle solidus surface topography. No melt is thus allowed to be emplaced into the crust outside of these chambers. Therefore, in the future, melt transport and emplacement algorithm need to be improved (e.g., by implementing more distributed melt emplacement via dikes).
(3) Magmatic accretion of the new oceanic crust. Spontaneous cooling and crystallization of melts are implemented at the walls of the lower-crustal magma regions[58]. The effective density of the molten crust in the magma region is calculated as

$$\rho_{\text{eff}} = \rho_{\text{solid}}\left(1 - M + M\frac{\rho_{0,\text{molten}}}{\rho_{0,\text{solid}}}\right), \tag{16}$$

where $\rho_{0,molten}$ and $\rho_{0,solid}$ are the standard densities of molten and solid oceanic crust, respectively.

Crystallization of magma and melting of the crust are computed from the simple linear batch melting model

$$M = \begin{cases} 0 & \text{for } T \leq T_{solidus} \\ (T - T_{solidus})/(T_{liquidus} - T_{solidus}) & \text{for } T_{solidus} < T < T_{liquidus} \text{ , } \quad (17) \\ 1 & \text{for } T \leq T_{liquidus} \end{cases}$$

The density of solid crust varies with pressure ($P$) and temperature ($T$) according to the equation,

$$\rho_{solid} = \rho_{0,solid}[1 - \alpha(T - 298)][1 + \beta(P - 0.1)], \quad (18)$$

where $\alpha = 3 \times 10^{-5}$ 1/K and $\beta = 10^{-5}$ 1/MPa are thermal expansion and compressibility of the crust. The effect of latent heating due to equilibrium crystallization of the crust from the magma regions is included implicitly by increasing the effective heat capacity and the thermal expansion of the partially crystallized/molten rocks[32].

(4) Hydrothermal circulation at the axis of the ridge. Rapid cooling of the new oceanic crust by hydrothermal circulation is parameterized with an enhanced thermal conductivity of the crust (50; 32).

$$\kappa_{eff} = \kappa + \kappa_0(Nu - 1)\exp\left(A\left(2 - \frac{T}{T_{max}} - \frac{y}{y_{max}}\right)\right), \quad (19)$$

$\kappa$ is the thermal conductivity (Table S2), $\kappa_0 = 3W/(m\cdot K)$ is the reference thermal conductivity, $Nu = 2$ is the assumed Nusselt number, $A = 0.75$ is a smoothing factor and $y$ is depth. $T_{max}$, 600 °C, is the cutoff maximum temperature. $y_{max}$, 6 km, is the cutoff maximum depth of hydrothermal circulation. Also, in order to ensure an efficient heat transfer from the upper surface of the plate in the Eulerian model, the thermal conductivity of the seawater layer above this plate is taken to be a hundred times higher (200 W/m/K) than that of the dry mantle (1–4 W/m/K).

**Model geometry, initial, and boundary conditions.** The initial model set-up is shown in Fig. S10. The onset of oceanic spreading is a young lithosphere with a 7 km thick crust. The Eulerian computational domain corresponds to a physical domain of $202 \times 202 \times 98$ km resolved with a regular grid of $405 \times 405 \times 197$ nodes. About 130 million Lagrangian markers are randomly distributed with the model domain. Free slip boundary conditions are imposed on the front and back boundaries. Spreading boundary conditions are constant spreading rate in x-direction ($v_{spreading} = v_{left} + v_{right}$, where $v_{left} = v_{right}$) and compensating vertical influx velocities through the upper and lower boundaries ($v_{top}$ and $v_{bottom}$) to ensure the conservation of mass. The free surface boundary condition is approximated on the top crust by a 10 km weak layer to simulate the topography. The symmetric initial thermal structure corresponds to a cooling age which linearly increases from 0.1 Myr in the center of the model to 10 Myr at the left and right boundaries. Constant temperature boundary conditions are used at the top and bottom of the model. 0 °C is implemented at the top, 1567 K at the bottom is set to mimic the mantle potential temperature at 1255 °C[59]. With this initially hot symmetrical ridge condition, we intended to avoid the influence of any initial ridge asymmetry for the model evolution. As the result, ridge asymmetry develops spontaneously from the initially symmetrical configuration upon cooling of the ridge toward the equilibrium thermal state (Fig. 3, S11–S15).

## Data availability

All the seismic tomography results presented in this article can be reproduced using the data and program codes, which are available in Zenodo[60]. The data and models on seismic tomography generated in this study are provided in the Supplementary Information file. All input and output files, used in the numerical modelling and visualization are available from the corresponding author upon request.

## Code availability

The LOTOS code used to calculate the seismic tomography model is available at Zenodo[60]. Researchers interested in using I3ELVIS code should contact T.V.G. (taras.gerya@erdw.ethz.ch).

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

## Acknowledgements
The work of I.K. and A.J. is supported by the Russian Science Foundation Grant #20-17-00075.

## Author contributions
I.K. was responsible for performing the tomography inversion and interpretation of the results. V.S. was one of the organizers of the field works and was responsible for the data processing. M.L. and T.G. created the numerical thermos-mechanical model. A.J. contributed in the adaptation of the tomography algorithm to the case of floating networks. A.I. was responsible for the analysis of the available geochemical data. All the co-authors participated in discussions of the results and the writing of the article.

## Competing interests
The authors declare no competing interests.
