## [Peer Review File · Nature Communications]

REVIEWER COMMENTS

Reviewer #1 (Remarks to the Author):

This is a useful paper that ties together previous volcanological observations, a new treatment of existing seismic data, and numerical modeling of melt generation and crustal construction.

The authors suggest all of these are consistent with one another, and in combination provide a new understanding of how it is possible to get explosive eruptions at super-slow spreading ridges. It's a nice story, reflects a lot of work, and therefore should be published after revision. As far as being worthy of a Nature journal, my answer is yes, because the combination of these three things is new to me. Maybe it is unique? So even if the results are not definitive, this is an important new approach and contribution. Future work can build upon it. Let's not hold it up too much by complaining about the details.

That said...the quality of the presentation can be improved, the English is poor in places, the understanding of the broader picture of oceanic volcanism has some misconceptions, the detailed interpretation of the seismic data in terms of lithologies present is surely uncertain, and the results of the modeling have aspects that are not consistent with some of what is observed along the Gakkel Ridge. None of these flaws are fatal, but if they were fixed I think a much improved paper would be possible. And in my view a more appropriate conclusion at the end would be to note that this is a first step with a lot of future potential.

There is one aspect that really needs to be dealt with, which comes from the geochemistry. The authors ignore the geochemical data that is available for the volcanic products of the explosive eruption they are modeling. CO₂ contents are known for these magmas, and the degassing depths can be accurately determined. They are not consistent with the model proposed, so the authors need to think how to modify the model so they do not violate the geochemical constraints, and they certainly need to reference and discuss the geochemical data that are directly pertinent.

My bottom line would be to recommend major revisions and a second round of review. There is a lot of interesting work that has been done here that should be in the literature.

I also would recommend that the authors need to be more generous to Sohn et al. Yes, the current work has a much more comprehensive approach, and the seismic and numerical results are far beyond what Sohn et al did. But the model they come up with is similar to what the earlier paper suggested. I know it is hard to write sentences like "our new interpretation of the seismic data coupled with numerical modeling leads to conclusions consistent with earlier suggestions." But that is the case here. The paper only references Sohn for the volcanological observations and inferred high CO₂ content. But Sohn et al. said:

"These results provide a new perspective for interpreting the 1999 seismic swarm and volcanic event at the 85° E site. The seismic swarm began with extensional events, but after three months the earthquakes changed to sources with large volume changes (implosions)⁶. Large-volume-change events are rare at MORs, but they are consistent with the rapid evacuation of explosive material from a deep-lying magma chamber. The sequence of extensional earthquakes leading up to the implosions may have perturbed the stress field enough to fracture the chamber roof,

thereby releasing pressurized magmatic volatiles. Rapid acceleration of decompressing volatiles may have triggered vulcanian explosions during the eruption³, consistent with the talus distribution observed on Oden volcano. Multiple episodes of explosive volatile discharge over a prolonged period are required for producing the variations in apparent age and thickness of the deposits we observed, and we note that small-magnitude explosive acoustic signals were detected by local (ice-mounted) seismic networks at the eruption site more than two years after the 1999 seismic swarm¹⁹. Explosive volatile discharge has clearly been a widespread, and ongoing, process at the 85° E segment.

Our results raise new questions about volatile processes in ultraslow-spreading magmatic systems. More observations will be necessary to determine the ubiquity of pyroclastic activity at ultraslow spreading rates (<15–20 mm yr⁻¹, full rate), but from first principles there is reason to believe that ultraslow-spreading ridges may be especially conducive to the build-up and explosive discharge of volatile-rich magmatic foams. Long time intervals between eruptions should increase the quantity of volatiles that can be accumulated in a magma chamber, and if the global correlation between spreading rate and magma chamber depth extends to ultraslow rates, then volatile build-up will occur deep within the crust at high storage pressures. Our results add to the growing body of evidence that ultraslow-spreading ridges host unique modes of crustal accretion and tectonic extension^{20,21}, and motivate continuing efforts to solve the technical and logistical issues that have impeded scientific access to these unique geological environments.”

This paper concludes: “Hence, we conclude that
170 ultraslow spreading ridges with low mantle potential temperature may not only focus deep
rising
171 partially molten asthenosphere towards magma chamber to form narrow magmatic sections,
but
172 also subsequently trigger volcanic eruptions in the rift valley due to the very high volatile
content
173 in the focused low-degree basaltic melts.”

Isn't that the same conclusion as Sohn et al?

Figure 4 of the present paper is consistent with this view, and the earlier paper also emphasizes that this is likely a consequence of super-slow spreading. So, the present authors have done a lot of novel work, but they might better acknowledge the previous contributions and ideas.

The language

Here is a statement from the first paragraph: *“The divergent processes along the spreading centers cause forming the new lithosphere,
32 which fully compensates for lithosphere submerging in subduction and collision zones. All
33 spreading zones are located in deep-water oceanic areas,”*

Or later:

Indeed, the P-wave velocity is more 104 sensitive to the composition and is normally higher in magmas of primitive composition arrived 105 from deeper sources (24).

This is a basaltic system, so p-wave velocity will not vary much within the small range of composition.

Or:

an ovoid magma chamber appears

168 with high temperature and high strain rate, which likely reveals strong fluid-melt activity and 169 further interprets the main source of volcanic eruptions occurring.

How can a magma chamber have a strain rate? It is a liquid. I have no idea what line 169 actually means.

There are many other examples throughout the paper. The numerical supplement seems to be well written. The authors with more facility in written English need to do a major edit to the paper, and also make sure the sentences make sense.

Ocean ridges

The authors make statements that are just wrong about ocean ridges. Not all ridges are submarine and deep (they are shallow where hot spots are nearby, Iceland is above sea level); there are quite a few segments as deep as the 85E volcano, ridge basalts are not generated by 15-20% melting. Here is the segment depth distribution from Gale et al. 85E is the big yellow symbol. Shallow ridge segments are off scale to the right.

These are mean depths. The actual 85E volcano depth is mostly shallower than 4000m. Super slow spreading ridges encompass a very large range in depth.

If numerical modeling is giving 15-20% as an average extent of melting, the models need to be revised. The maximum extent of melting gets that high, but mean melt production is approximately the crustal thickness divided by the maximum dry melting depth, corrected for density. Since melting starts at 60km or more, if you melt even 15% you get about 10km of

lower density crust. Ocean crust is normally 6km thick. Extents of melting are mostly 8-15%. They cannot be 2% beneath the Gakkel. In that case you would never get crust more than a couple of hundred meters thick.

Seismology

I am not a seismologist, so not qualified to comment on the detailed seismic treatment. But from the outsider's point of view, the interpretation seems inherently problematic because no S waves are received by seismometers sitting on top of the ocean. For this reason the authors have to depend on the P-S wave conversion. While real S-waves can be picked reliably, it is not so clear from the examples given in the appendix that P-S conversion picks are as reliable. The testing they do does not take this uncertainty into account. Furthermore, converting the V_p/V_s "inferred V_s " to melt fraction and detailed lithology is non-unique and quite unconstrained. I hope a seismology expert is engaged to give views on these points.

Modeling

I am not a numerical modeler. The appendix gives a lot of detail about the numerical choices that were made, but some aspects are strange and raise questions. First, the initial condition shown in Figure S6 of a constant 7km crust just does not apply in this region. Do model results depend on that initial condition? If you start with the run after 10ma as the initial condition, what happens? Does that control the faulting that eventually occurs? Second, the 85E volcano is an isolated edifice surrounded by 100km on either side with no apparent volcanism. The modeling results, however, lead to 20km spacing between upwellings. So there appears to be a first order misfit. Third, is there clear geological evidence for a detachment fault in this region (e.g. unroofed lower crust, straited bathymetry, assymmetric cross sections)? And yet such a fault seems important for the modeling results and interpretation in the paper. Fourth, is the choice of 10.4 million years made because it leads to the desired results? What is happening at other times?

Could there be a movie showing the evolution from starting conditions to >10.2 Ma so the reader can see what actually happens, how variable the results are, and what the pertinent time scales are? I would love to see it.

Geochemistry

As a geochemist, I think the model needs modification in light of the geochemical constraints. We know that the CO_2/Ba ratio is constant in MORB, with a value of about 80 (See le Voyer et al, 2019 G-cubed on MORB primary magma CO_2 contents). Gakkel basalts from this region have about 20ppm Ba, so that suggests CO_2 contents of 1600ppm. Indeed, that is consistent with the melt inclusion values determined for the lavas from the explosive eruptions by Shaw et al. (EPSL Volume 289, Issues 3–4, 31 January 2010, Pages 311-322). Solubility pressures for a magma with the CO_2 and H_2O contents of these magmas can be determined with VolatileCalc (<https://www.usgs.gov/software/volatilecalc>). Solubility pressure is 3kb, which is the pressure just above the very top of the proposed magma reservoir in Figure 4. No magma in the reservoir shown will degas. OK, move the reservoir up a little? No, 3kb is the solubility pressure, and you have to degas from 3-2kb in order to release 40% of the CO_2 as gas. You can also calculate

degassing paths in VolatileCalc. To get the 14% CO₂ the authors propose for an explosive eruption, that gas needs to accumulate at some shallower level, and to have accumulated from a volume of magma that is more than one hundred times the erupted volume, because you do not completely degas the magma. It is also not obvious to me how you trap it. These volumetric constraints also mean that these kinds of eruptions must be rare—they can only be one percent of magma production. That makes generalizations about explosivity and super slow spreading rates a bit difficult.

In any case, the authors cannot avoid the CO₂ constraints. There cannot be a deep magma chamber that is the source of the eruptions, because no degassing occurs there. There are also strict limits on the amount of CO₂ that can be in the primary magma—you do not get just to gather CO₂ from large volumes of asthenosphere and not also gather Ba and Th. And the Ba contents of the erupted rocks are the same as the Gale et al. Average MORB, so the CO₂ contents are not particularly elevated, and it is not exceptionally high CO₂ in the primary magmas that is the ultimate cause. It also means the source of the eruptions needs to be a holding tank much shallower in the crust. Geochemical conclusions, like what the extent of melting is and how much CO₂ there is, are constrained by the geochemical observations from the erupted magmas.

So in view of these considerations the conclusion on lines 128-133 of a magma reservoir that is the source of the eruptions at 10-15km is simply not possible.

One might think in view of these issues that I would suggest the paper be rejected. But that is not my view, because the paper reflects a lot of work and a novel approach that has promise. That alone makes it a worthwhile contribution to the literature. Further work of this kind could make a major contribution to our understanding of ocean ridges.

Charles Langmuir
Harvard University

Reviewer #2 (Remarks to the Author):

Review of "Low-degree mantle melting controls deep seismicity and explosive volcanism of the Gakkel Ridge"

This manuscript presents the results from coupled seismic and thermo-mechanical modeling analysis carried out beneath the Gakkel Ridge. Some of the data/results build on previous studies, i.e., seismicity, but both the 3D velocity and the thermo-mechanical models are new and shed light on some fundamental processes of magmatic/tectonic accretion at ridges. Both results and interpretation leading to providing a plausible scenario for what causes explosive volcanism and deep seismicity beneath the Gakkel ridge would be of interest for the marine geodynamics and ridge communities at large. Overall, the paper is well-written and the authors make a good job at presenting and discussing their results. I have a few comments though, detailed below, that need clarification before any publication.

My detailed comments in order are as follows:

In main manuscript:

Line 46: the authors should consider adding more recent papers published in the last couple of years, as for example:

Antonovskaya, G. N., Kapustian, N. K., Konechnaya, Y. V., & Danilov, A. V. (2020). Registration capabilities of Russian Island-based seismic stations: Case study of the Gakkel ridge monitoring. *Seismic Instruments*, 56(1), 33-45.

Lines 49-50: is it really a frequent feature? I would tone down on the "frequent" and remove it. Some studies show otherwise, i.e., < 18 km depth, e.g., Grevemeyer, I., et al., 2019.

Line 57: change "contaminated" by "enriched"

Line 60: Why is the authors stating that seismicity reaches 25 km depth when they show on figure 2 (red dots) that some EQs were recorded at 32 km focal depth (which is impressively deep and not discussed)?

Lines 114-120: this seismic gap inferred and interpreted as the consequence of crustal rocks being too soft (surface: more fractured and saturated with seawater and in depth: remnant magma pocket), which would prevent accumulation of stresses and generation of seismicity, have been observed in Schlindwein and Schmid's (2016) study of SWIR micro-seismicity. Later, Grevmeyer et al. (2019) analysis of the SWIR dataset indicates that the previously proposed thick aseismic region in the upper lithosphere was "simply a consequence of a model embedded in the hypocenter location procedure that did not include a few 100 m-thickness of

unconsolidated sediment". They show the strong influence of such layer (and hence the importance of including such layer in models) on the focal depth and apparent seismic gap. Furthermore, because of their very slow spreading rate, ultraslow-ridges are known to have substantial sediment thick layer at the axis and flanks.

Was any unconsolidated sediment layer considered in this present study to avoid this bias which would result in having artifacts, i.e., seismic gap and deep seismicity? If not, how can you state that this isn't a bias of not considering such layer and can you provide model results including such layer for comparison?

Line 125: here again, a depth of about 25 km is mentioned while they clearly show deeper seismicity (up to 32 km) – is the 32 km EQ(s) considered an outlier(s)? If so, it needs to be explicated and discussed.

Line 127: this is a highly speculative statement. I would rephrase it: "We speculate a fault crossing the entire lithosphere".

While Tao et al., 2020 paper focuses more on the coupling between seismicity and hydrothermalism, some of their results aligned with this paper findings should be acknowledged here, i.e., showing a deep detachment fault reaching 15 km deep associated to deep seismicity, and asymmetry.

Tao, Chunhui, W. E. Seyfried, R. P. Lowell, Yunlong Liu, Jin Liang, Zhikui Guo, Kang Ding et al. "Deep high-temperature hydrothermal circulation in a detachment faulting system on the ultra-slow spreading ridge." Nature communications 11, no. 1 (2020): 1-9.

Line 137: what does "self-consistent" means here?

Line 137: is it not thermo-mechanical numerical modelling to explore magmatic processes rather than a magmatic-thermo-mechanical numerical modelling? See your line 178.

Lines 150-151: some kind of pleonasm here - by definition if a section is amagmatic then the magmatic activity is rare. Perhaps reformulate this sentence to avoid the redundancy.

Lines 151-152: is it the spreading ridgeS as in several ridgeS succeeding each other or is it one ridge (Gakkel) with several segments/sections that are not dissected by transform fault? It needs clarification.

Line 183: replace "which corresponds" with "which would correspond"

Line 187: remove "outstanding". Deep seismicity and asymmetry are common to all ultra-slow spreading ridges (including the Mohn and Knipovitch ridges) as stated by the authors in the introduction who referenced all previous work done at these ridges.

Lines 196-197: this is inaccurate. Define "regular spreading zones". The statement saying that "magma generation occurs continuously along ridges" is only true for fast-spreading ridges which are not accounting for all the "regular spreading zones". Both slow-spreading and ultra-

slow spreading ridges have discrete magmatic zones along segments/sections. It is not a specific characteristic of the Gakkel per se but rather a shared characteristic with the others ultra-slow and slow-spreading ridges to some extent. Please refer to Cannat's 1993 paper and all the literature that followed. Please rephrase your statement.

Cannat, M. (1993). Emplacement of mantle rocks in the seafloor at mid-ocean ridges. *Journal of Geophysical Research: Solid Earth*, 98(B3), 4163-4172.

In Supporting Information:

Lines 9-11: This statement is not quite accurate. Spreading rate for Mohn and Knipovitch ridges is about 15-17 mm.a⁻¹ (Bruvoll et al., 2009), which is < 20mm.a⁻¹, hence these two ridges belong as well to the ultraslow spreading ridges. Please reword.

Lines 17-19: here again this statement is not quite accurate. Gakkel is not the only one not being dissected by transform faults - both Mohn and Knipovitch ridges share this "difference". Please reword.

Reviewer #3 (Remarks to the Author):

The manuscript tackles a major unsolved issue of mid-ocean ridge research, i.e., the relationship between low-degree of mantle melting and the occurrence of explosive volcanism along the ultraslow spreading Gakkel Ridge in the Arctic Ocean. The authors provide constraints from seismic tomography - revealing structural elements and numerical modelling - indication/simulating potential processes. In concert, both may provide indeed a new perspective on the mechanisms controlling the rather puzzling behaviour of ultraslow spreading ridges, which - for a long time - were considered to be rather amagmatic and therefore scientists were rather surprised to find explosive volcanism and patches of robust magmatic accretion. In general, I therefore believe that the manuscript can make an important contribution.

However, there are a number of features that need to be improved before the manuscript can be finally evaluated. First, the text and title are still rather "sloppy". For example, the title highlights the features to be discussed in the paper, but overall, it's nothing new. Thus, rocks showing a low degree of mantle melting were sampled years ago. And of course - the mantle melting is the underlying process that controls features occurring at shallower level - like deep seismicity and occurrence of explosive volcanism. The fact that seismicity shows a very deep level of activity was shown by one of the authors previously (Korger and Schlindwein, *Geophys. J. Int.*, 2012) and also the occurrence of explosive volcanism is well known (Sohn et al., *Nature*, 2003). When reading the manuscript, I'm lost if we really learn something new. Therefore, the authors fail to make their point clearly. However, some of my personal uncertainty about the manuscript comes from the style of writing and how terms are chosen/used. Thus, at several places non-unique geophysical observations from the tomography are presented as supporting a unique explanation. For example, high Vp/Vs ratios are presented as being a solid fact for melting. Of course - at settings where melts migrate upwards geophysical data often reveal

high V_p/V_s ratios. However, also settings with a high degree of serpentinization show high V_p/V_s ratios (Christensen, 2004) and serpentinization is a common process at ultraslow spreading ridges. Further, a fractured rock with fluid filled veins also has a high V_p/V_s ratio (Popp & Kern, 1994; Wang et al., 2012). Thus, it's rather misleading to claim that V_p/V_s provides unique evidence for occurrence of melting – at least in the lithosphere of an ultraslow spreading mid-ocean ridge. However, the numerical simulations may suggest that at the depth/location where the tomography finds high V_p/V_s ratios, melts should occur. In this case, the inter-fingering of simulations and evidence from the seismic tomography would support a new point of view that may indeed merit publication. Unfortunately, the current manuscript isn't making the point – or if it does, it's not well presented. Thus, to really make the paper a manuscript of interest to a broad audience, results from the seismic tomography should be linked in the discussion much more clearly to the features of the numerical model. Further, observation should be presented in a way that alternative interpretations are not ignored (like high V_p/V_s could be either melts, serpentinized rock or fractured and fluid filled rocks – or a combination of all).

Further, the relationship between features found in the seismic tomography and the numerical simulations need to be explored in more detail. In the current version of the manuscript I have sometimes the feeling that the comparison of features is rather incidental (i.e., comparing only one or two depth slices instead of using the geometry of a 3D volume). Thus, to relate the pattern found in the simulations to features found in the tomography, you only used one depth slice at 20 km. What will be needed is to show it in 3D.

In summary, I do see a paper with interesting and new results from the Gakkel Ridge, providing great structural images from seismic tomography. Further, numerical simulations are used to yield (some of) the processes controlling the behaviour of mantle melting and thus both datasets together can provide a new and deeper understanding. However, the current paper fails to do it. First, in the introduction the open questions and mysteries of the Gakkel Ridge should be clearly stated, to help readers to understand the aim of the study and why the different approaches were used and why they nurture each other. Right now, the aim of the study isn't really stated. Second, results from the tomography should be discussed more openly. Do not try to "hide" that some interpretations might not be unique. Third, describe more clearly the features arising from the simulations. To me it's unclear if melt migration is really a part of the output of the modelling or if it is approximated from the other parameters (note - I'm not a modeller myself and thus it should be clear to "dummies"). Last, discuss both approaches in concert and thus show readers why your results are so cool to merit publication in Nature Communication. What's the ground-breaking result? As I iterated above – all features from the title have been known before. So – why your study and what's the benefit?

Additional comments (mostly taken from the annotated m/s; please note the m/s has more comments and suggestions and I only copied a few into this formal review):

1.) Some of the SP waves in Supplementary Figure 1 show SP waves which have larger amplitudes than P - as expected. However, some have smaller amplitudes. Are those really SP waves or are they some P-wave multiples of sidescapes and not converted S-wave energy?

Please provide Wadati plots (or at least one plot combining P-time vs. S-time of all pairs) in the supplement. A Wadati plot of each earthquake should help to define better if picks are indeed appropriate or spurious.

2.) Please also show histograms of residuals of both P- and SP. The quality of tomographic results is rather poorly presented.

3.) Korger and Schlindwein (GJI, 2014) provided a P-wave tomography from the very same data used in this study. I have to say that the transects from both studies are rather difficult to compare as the strike of depth sections is slightly different. However, Korger and Schlindwein didn't reveal any P-wave anomaly where this study has a strong positive velocity anomaly. Which tomography shall I trust? Rather puzzling and I would like to get a good explanation for the different features stemming from the same data. Considering that travel time data are the same I cannot see why the models should be so much different.

4.) Why occur some Vp/Vs anomalies with a similar amplitude away from the proposed magmatic conduit (to the north and west)? If it is really magmatic activity causing the anomaly and the activity is linked to the local earthquakes - why have the other anomalies no quakes? Or are these features not so well resolved and hence artefacts? Based on Korger and Schlindwein only the central area had a large number of rays and good coverage. How much bias is caused by rays having longer offsets. For those, the tomography will have a rather low resolution at shallow depth. However, shallow features are important and show generally the largest lateral variations, causing significant delays. How robust is the inversion of deep-seated features when the shallow structure lacks resolution/coverage?

5.) Please discuss what is the origin of the other Vp/Vs anomalies? Melts, too? Fracturing?

6.) The swath mapping had a rather limited coverage – only at the centre of the study area. How was the seafloor depth constrained where no swath data exists? From IBCAO2 bathymetry? In this case ray entry points into the seafloor might be biased as the seafloor is often based on interpolated data and not measured soundings, in turn, causing significant delays and hence limiting the resolution power of the tomography.

7.) In the supplement it is described that the water layer is approximated by a constant velocity. However, the water in the arctic is strongly layered (water mass with very little salinity at the top, Atlantic bottom water at greater depth which a much higher salinity and possible some mixing in-between). I guess that layering of the water layer will have some effect on the location of ray entry points into the seafloor. Considering the large variations of the seafloor this should cause some bias. Why not considering the real layering of seawater in the tomographic model?

8.) Seismicity in the sections is plotted without error bars and interpreted as being very precise (discussed in terms of indicating a fault zone). However, even in a perfect Earth travel times are associated with some uncertainty when picking onsets in noisy data. First, please report the magnitude of errors for P- and S-wave onsets (at least in the supplement). Further, how large is the horizontal and vertical uncertainty? Please plot error bars from the location procedure. At least indicate the average uncertainty (as is often done in similar studies).

9.) Line 139-141: "The modeling results show that the ultraslow spreading rate and low mantle potential temperature are critical to the formation of a very low magma supply by volatile-rich low-degree melts (7,8) derived from the slowly rising and decompressing asthenosphere." – awkward sentence. Either this are your results (in this case you do not need a reference to any paper) or it's from the cited papers. In this case it won't be your results. This is a good example of what I considered "sloppy language", which leaves reader wondering about the results. If

your study and the other paper(s) show the same – just state it like that.

10.) The numerical modelling results are difficult to place. Here, only results from ultraslow spreading rates are presented. If the findings are so unique to ultraslow spreading ridges, your case would be much stronger when you show that it's not occurring at slightly faster spreading rates and it would also be nice to learn at which spreading rate the transition from one mode to another occurs.

11.) I added a comment to the m/s that the process that leads to the accumulation of volatiles needs to be explored in much more details. I believe that this is really one of the most important features, highlighting the link between ultraslow seafloor spreading and explosive magmatism. Right now, it's just one of many features in the m/s and not well exposed.

12.) The numerical model reveals very thick crust – up to 20 km thick. This is not consistent with observations showing very thin crust in the Arctic (Jokat & Schmid-Aursch, GJI, 2007) and the conceptual model presented in Fig. 4. Please explain. Or is the numerical model using a different definition of crust? However, this is rather awkward.

I added in total 69 comments (some just minor grammatical features) to the annotated m/s. Please consider those when revising the manuscript. I believe that the content of the m/s is of great interest and will have a considerable impact – when presented differently. This is perhaps just my personal point of view and I might be mistaken, driven by my ignorance. However, when I have difficulties in following the m/s – I'm afraid that others will have similar difficulties, too.

Good luck with the revision, Ingo Grevemeyer

Additional references not in the manuscript but cited in my review:

Christensen, N. I. (2004), *Serpentinites, peridotites, and seismology*, *Int. Geol. Rev.*, 46, 795–816, doi:10.2747/0020-6814.46.9.795.

Popp, T. and H. Kern (1994), *The influence of dry and water saturated cracks on seismic velocities of crustal rocks – A comparison of experimental data with theoretical models*, *Surv. Geophys.*, 15, 443-465.

Wang, X.-Q., A. Schubnel, J. Fortin, E. C. David, Y. Guéguen, and H.-K. Ge (2012), *High Vp/Vs ratio: Saturated cracks or anisotropy effects?*, *Geophys. Res. Lett.*, 39, L11307, doi:10.1029/2012GL051742.

Rebuttal letter on the revised paper by Ivan Koulakov, Vera Schlindwein, Mingqi Liu, Taras Gerya, Andrey Jakovlev and Alexei Ivanov “Low-degree mantle melting controls deep seismicity and explosive volcanism of the Gakkell Ridge” submitted to Nature Communications (Paper # NCOMMS-21-38084-T)

The author’s responses are indicated with “**REP:**” and highlighted with the red color. Line numbering corresponds to the version of the manuscript with tracked changes.

REVIEWER COMMENTS

Reviewer 1

This is a useful paper that ties together previous volcanological observations, a new treatment of existing seismic data, and numerical modeling of melt generation and crustal construction.

The authors suggest all of these are consistent with one another, and in combination provide a new understanding of how it is possible to get explosive eruptions at super-slow spreading ridges. It’s a nice story, reflects a lot of work, and therefore should be published after revision. As far as being worthy of a Nature journal, my answer is yes, because the combination of these three things is new to me. Maybe it is unique? So even if the results are not definitive, this is an important new approach and contribution. Future work can build upon it. Let’s not hold it up too much by complaining about the details.

That said...the quality of the presentation can be improved, the English is poor in places, the understanding of the broader picture of oceanic volcanism has some misconceptions, the detailed interpretation of the seismic data in terms of lithologies present is surely uncertain, and the results of the modeling have aspects that are not consistent with some of what is observed along the Gakkell Ridge. None of these flaws are fatal, but if they were fixed I think a much improved paper would be possible. And in my view a more appropriate conclusion at the end would be to note that this is a first step with a lot of future potential.

REP: We did our best to follow these recommendations (see our responses below).

There is one aspect that really needs to be dealt with, which comes from the geochemistry. The authors ignore the geochemical data that is available for the volcanic products of the explosive eruption they are modeling. CO₂ contents are known for these magmas, and the degassing depths can be accurately determined. They are not consistent with the model proposed, so the authors need to think how to modify the model so they do not violate the geochemical constraints, and they certainly need to reference and discuss the geochemical data that are directly pertinent.

REP: Our new co-author Alexei Ivanov took care of the geochemical issues and made an important contribution to address the problems mentioned by the reviewer (see our responses below).

My bottom line would be to recommend major revisions and a second round of review. There is a lot of interesting work that has been done here that should be in the literature. I also would recommend that the authors need to be more generous to Sohn et al. Yes, the current work has a much more comprehensive approach, and the seismic and numerical results are far beyond what Sohn et al did. But the model they come up with is similar to what the earlier paper suggested. I know it is hard to write sentences like “our new interpretation of the seismic data coupled with numerical modeling leads to conclusions consistent with earlier suggestions.” But that is the case here. The paper only references Sohn for the volcanological observations and inferred high CO₂ content. But Sohn et al. said:

“These results provide a new perspective for interpreting the 1999 seismic swarm and volcanic event at the 85° E site. The seismic swarm began with extensional events, but after three months the earthquakes changed to sources with large volume changes (implosions)⁶. Large-volume change events are rare at MORs, but they are consistent with the rapid evacuation of explosive material from a deep-lying magma chamber. The sequence of extensional earthquakes leading up to the implosions may have perturbed the stress field enough to fracture the chamber roof, thereby releasing pressurized magmatic volatiles. Rapid acceleration of decompressing volatiles may have triggered vulcanian explosions during the eruption³, consistent with the talus distribution observed on Oden volcano. Multiple episodes of explosive volatile discharge over a prolonged period are required for producing the variations in apparent age and thickness of the deposits we observed, and we note that small-magnitude explosive acoustic signals were detected by local (ice-mounted) seismic networks at the eruption site more than two years after the 1999 seismic swarm¹⁹. Explosive volatile discharge has clearly been a widespread, and ongoing, process at the 85° E segment.

Our results raise new questions about volatile processes in ultraslow-spreading magmatic systems. More observations will be necessary to determine the ubiquity of pyroclastic activity at ultraslow spreading rates (<15–20 mm yr⁻¹, full rate), but from first principles there is reason to believe that ultraslow-spreading ridges may be especially conducive to the build-up and explosive discharge of volatile-rich magmatic foams. Long time intervals between eruptions should increase the quantity of volatiles that can be accumulated in a magma chamber, and if the global correlation between spreading rate and magma chamber depth extends to ultraslow rates, then volatile build-up will occur deep within the crust at high storage pressures. Our results add to the growing body of evidence that ultraslow-spreading ridges host unique modes of crustal accretion and tectonic extension^{20,21}, and motivate continuing efforts to solve the technical and logistical issues that have impeded scientific access to these unique geological environments.”

This paper concludes: “Hence, we conclude that ultralow spreading ridges with low mantle potential temperature may not only **focus** deep rising partially molten asthenosphere towards magma chamber **to form narrow magmatic sections**, but also subsequently trigger volcanic eruptions in the rift valley due to the very high volatile content in the focused low-degree basaltic melts.”

Isn't that the same conclusion as Sohn et al?

Figure 4 of the present paper is consistent with this view, and the earlier paper also emphasizes that this is likely a consequence of super-slow spreading. So, the present authors have done a lot of novel work, but they might better acknowledge the previous contributions and ideas.

REP: We agree that in many aspects our results are consistent with the concept proposed by Sohn and others. We have highlighted this in some parts of the text (for example in L207-208). However, we slightly disagree that our conclusions completely repeat the statements proposed by Sohn et al. (2008). In the highlighted pieces of text from the Sohn's work, there is discussion about a large amount of volatiles responsible for explosive volcanism in the ultraslow spreading settings. In our work, we do not investigate the volatiles, but mostly use the information about them provided by other authors, including Sohn et al. At the same time, as far as we aware, the previous studies did not find a definitive answer to a question on where this high concentration of volatiles comes from and why such centres of activity focus in narrow sections. Based on seismic tomography, we show for the first time the location of the magma below the spreading centre, and our numerical model provides an explanation of focusing such volatile-rich magma reservoirs in the conditions of relatively cold spreading zone. As far as we know, none of the existing articles considered the origin of volcanism in the Gakkel Ridge in this aspect.

The language

Here is a statement from the first paragraph: “The divergent processes along the spreading centers cause forming the new lithosphere, which fully compensates for lithosphere submerging in subduction and collision zones. All spreading zones are located in deep-water oceanic areas,”

REP: We have replaced “All spreading zones” with “Most of spreading centres”. For the first part of the highlighted phrase, we do not understand what is wrong.

Or later:

104 Indeed, the P-wave velocity is more sensitive to the composition and is normally higher in magmas of primitive composition arrived from deeper sources (24).

This is a basaltic system, so p-wave velocity will not vary much within the small range of composition.

REP: We have replaced this statement with: “Indeed, the P-wave velocity is more sensitive to the composition and is normally higher in magmas arrived from the mantle than in crustal rocks (24)” (L116-117).

Or:

168 an ovoid magma chamber appears with high temperature and high strain rate, which likely reveals strong fluid-melt activity and further interprets the main source of volcanic eruptions occurring.

How can a magma chamber have a strain rate? It is a liquid. I have no idea what line 169 actually means.

REP: In our numerical model, the lower crustal magma chamber is only partially molten and contains crystal mush rather than pure liquid. We removed mention of strain rate to avoid confusion.

There are many other examples throughout the paper. The numerical supplement seems to be well written. The authors with more facility in written English need to do a major edit to the paper, and also make sure the sentences make sense.

REP: The English language of our manuscript was checked by professional editors from American Journal Experts.

Ocean ridges

The authors make statements that are just wrong about ocean ridges. Not all ridges are submarine and deep (they are shallow where hot spots are nearby, Iceland is above sea level); there are quite a few segments as deep as the 85E volcano, ridge basalts are not generated by 15-20% melting. Here is the segment depth distribution from Gale et al. 85E is the big yellow symbol. Shallow ridge segments are off scale to the right.

These are mean depths. The actual 85E volcano depth is mostly shallower than 4000m. Super slow spreading ridges encompass a very large range in depth.

REP: In L59, we replaced “at depths of more than 4000 meters” with “at depths of 3700-4000 meters”, which gives more accurate range for the bottom depth in the volcanic area.

If numerical modeling is giving 15-20% as an average extent of melting, the models need to be revised. The maximum extent of melting gets that high, but mean melt production is approximately the crustal thickness divided by the maximum dry melting depth, corrected for density. Since melting starts at 60km or more, if you melt even 15% you get about 10km of lower density crust. Ocean crust is normally 6km thick. Extents of melting are mostly 8-15%.

They cannot be 2% beneath the Gakkel. In that case you would never get crust more than a couple of hundred meters thick.

REP: The extent of melting is clarified. In our numerical model, the average extent of melting is about 9-11% (please see the figure below). Higher degree of melting is only characteristic for the model initiation due to the prescribed initial hot temperature profile along the ridge (Fig. S16), which cools down with time and goes toward the colder equilibrium state (Fig. S17).

Distribution of melting degree on mantle markers at both magmatic and amagmatic sections. (a) The extent of melting at magmatic section. (b) The extent of melting at amagmatic section. (c) The extent of melting along the dashed white lines in (a) and (b). The very high extent of melting (red color) in (a) and (b) is caused by the initial thermal configuration.

Seismology

I am not a seismologist, so not qualified to comment on the detailed seismic treatment. But from the outsider's point of view, the interpretation seems inherently problematic because no S waves are received by seismometers sitting on top of the ocean. For this reason the authors have to depend on the P-S wave conversion. While real S-waves can be picked reliably, it is not so clear from the examples given in the appendix that P-S conversion picks are as reliable. The testing they do does not take this uncertainty into account. Furthermore, converting the V_p /"inferred V_s " to melt fraction and detailed lithology is non-unique and quite unconstrained. I hope a seismology expert is engaged to give views on these points.

REP: We have added a paragraph, in which we describe in details how the manual picking of the direct P and converted P-S waves was conducted and how the correctness of the phase identification was tested (L107-123 of Supplementary).

Regarding to the non-uniqueness of interpretation of the derived seismic model in terms of petrological parameters, we admit that this problem cannot be solved quantitatively. However, we note that an interpretation can be proposed based on similarity of seismic structures obtained for a number of other active volcanoes, where multidisciplinary studies provided robust information about magma plumbing systems. We have added a paragraph in the main paper summarizing this issue (L 105-113 of the main paper).

Modeling

I am not a numerical modeler. The appendix gives a lot of detail about the numerical choices that were made, but some aspects are strange and raise questions. First, the initial condition shown in Figure S6 of a constant 7km crust just does not apply in this region. Do model results depend on that initial condition? If you start with the run after 10ma as the initial condition, what happens? Does that control the faulting that eventually occurs?

REP: Explanation of the initial conditions has been extended. With this initially hot symmetrical ridge condition we intended to avoid influence of any initial ridge asymmetry for the model evolution. As the result, ridge asymmetry develops spontaneously from the initially symmetrical configuration upon cooling of the ridge toward the equilibrium thermal state. At the beginning, the initial condition (e.g., thermal configuration and oceanic crust) can indeed affect the model results in term of elevated degree of mantle melting. However, after about 3 Myr, the effect of initial setup in model results becomes negligible.

Second, the 85E volcano is an isolated edifice surrounded by 100km on either side with no apparent volcanism. The modeling results, however, lead to 20km spacing between upwellings. So there appears to be a first order misfit.

REP: The discussion of modelling results has been extended. The reference model shows about 20 km spacing between upwelling, which is indeed not consistent with the observation. However, through exploring the effect of different spreading rates and mantle potential temperature for the distribution of magmatic segments, we found that reduced spreading rate and lower mantle potential temperature can increase the spacing between upwellings. Thus, if the reduced spreading rate and lower mantle potential temperature are implemented, the larger spacing can be produced. This however does not change the discussed relationship and structure of magmatic vs. amagmatic segments.

Third, is there clear geological evidence for a detachment fault in this region (e.g. unroofed lower crust, straited bathymetry, assymmetric cross sections)? And yet such a fault seems important for the modeling results and interpretation in the paper.

REP: This region shares consistent axial morphology and depth with Mid-Atlantic ridge and Southwest Indian ridge where detachment faults dominate at spreading centers (Pontbriand et al., 2012). In addition, according to the study in the slow-spreading Eurasia Basin, Arctic Ocean, detachment faults that expose rocks from deep below this region are proposed (Lutz et al., 2018).

Pontbriand, C. W., Soule, S. A., Sohn, R. A., Humphris, S. E., Kunz, C., Singh, H., ... & Shank, T. (2012). Effusive and explosive volcanism on the ultraslow- spreading Gakkel Ridge, 85 E. *Geochemistry, Geophysics, Geosystems*, 13(10).

Lutz, R., Franke, D., Berglar, K., Heyde, I., Schreckenberger, B., Klitzke, P., & Geissler, W. H. (2018). Evidence for mantle exhumation since the early evolution of the slow-spreading Gakkel Ridge, Arctic Ocean. *Journal of Geodynamics*, 118, 154-165.

Fourth, is the choice of 10.4 million years made because it leads to the desired results? What is happening at other times?

Could there be a movie showing the evolution from starting conditions to >10.2 Ma so the reader can see what actually happens, how variable the results are, and what the pertinent time scales are? I would love to see it.

REP: We added two movies for the model evolution. The model ran till 12 Myr. The result near the end (10.4 Myr) was shown in the paper. The evolution from starting conditions to 12 Myr are shown in the movie. In the movie, "Gakkel_slice.avi", composition in 3D is shown with transparent solid mantle. Evolution of vertical slice

along $z = 63.5$ km and horizontal slice along $y = 20$ km are also shown in the movie. In the other movie, “Gakkel_surface.avi”, crustal thickness, thermal gradient, topography, melt depth and the configuration along the ridge are shown.

Geochemistry

As a geochemist, I think the model needs modification in light of the geochemical constraints. We know that the CO₂/Ba ratio is constant in MORB, with a value of about 80 (See le Voyer et al, 2019 G-cubed on MORB primary magma CO₂ contents). Gakkel basalts from this region have about 20ppm Ba, so that suggests CO₂ contents of 1600ppm. Indeed, that is consistent with the melt inclusion values determined for the lavas from the explosive eruptions by Shaw et al. (EPSL Volume 289, Issues 3–4, 31 January 2010, Pages 311-322). Solubility pressures for a magma with the CO₂ and H₂O contents of these magmas can be determined with VolatileCalc (<https://www.usgs.gov/software/volatilecalc>). Solubility pressure is 3kb, which is the pressure just above the very top of the proposed magma reservoir in Figure 4. No magma in the reservoir shown will degas. OK, move the reservoir up a little? No, 3kb is the solubility pressure, and you have to degas from 3-2kb in order to release 40% of the CO₂ as gas. You can also calculate degassing paths in VolatileCalc. To get the 14% CO₂ the authors propose for an explosive eruption, that gas needs to accumulate at some shallower level, and to have accumulated from a volume of magma that is more than one hundred times the erupted volume, because you do not completely degas the magma. It is also not obvious to me how you trap it. These volumetric constraints also mean that these kinds of eruptions must be rare—they can only be one percent of magma production. That makes generalizations about explosivity and super slow spreading rates a bit difficult.

In any case, the authors cannot avoid the CO₂ constraints. There cannot be a deep magma chamber that is the source of the eruptions, because no degassing occurs there. There are also strict limits on the amount of CO₂ that can be in the primary magma—you do not get just to gather CO₂ from large volumes of asthenosphere and not also gather Ba and Th. And the Ba contents of the erupted rocks are the same as the Gale et al. Average MORB, so the CO₂ contents are not particularly elevated, and it is not exceptionally high CO₂ in the primary magmas that is the ultimate cause. It also means the source of the eruptions needs to be a holding tank much shallower in the crust. Geochemical conclusions, like what the extent of melting is and how much CO₂ there is, are constrained by the geochemical observations from the erupted magmas.

REP: This comment is very useful and allowed us to modify the model in accordance with the published geochemical data of Shaw et al. (2010). Taking CO₂ concentrations of 1600 ppm and VolatileCalc we obtain 3.1 kbar saturation pressure which corresponds to the depth of anomaly 2. We have added a paragraph in L260-276 in the main paper and modified Fig. 4 (indicating much shallower depths of degassing).

So in view of these considerations the conclusion on lines 128-133 of a magma reservoir that is the source of the eruptions at 10-15km is simply not possible.

One might think in view of these issues that I would suggest the paper be rejected. But that is not my view, because the paper reflects a lot of work and a novel approach that has promise.

That alone makes it a worthwhile contribution to the literature. Further work of this kind could make a major contribution to our understanding of ocean ridges.

Charles Langmuir

Harvard University

Reviewer #2 (Remarks to the Author):

Review of “Low-degree mantle melting controls deep seismicity and explosive volcanism of the Gakkel Ridge”

This manuscript presents the results from coupled seismic and thermo-mechanical modeling analysis carried out beneath the Gakkel Ridge. Some of the data/results build on previous studies, i.e., seismicity, but both the 3D velocity and the thermo-mechanical models are new and shed light on some fundamental processes of magmatic/tectonic accretion at ridges. Both results and interpretation leading to providing a plausible scenario for what causes explosive volcanism and deep seismicity beneath the Gakkel ridge would be of interest for the marine geodynamics and ridge communities at large. Overall, the paper is well-written and the authors make a good job at presenting and discussing their results. I have a few comments though, detailed below, that need clarification before any publication.

My detailed comments in order are as follows:

In main manuscript:

Line 46: the authors should consider adding more recent papers published in the last couple of years, as for example:

Antonovskaya, G. N., Kapustian, N. K., Konechnaya, Y. V., & Danilov, A. V. (2020).

Registration capabilities of Russian Island-based seismic stations: Case study of the Gakkel ridge monitoring. *Seismic Instruments*, 56(1), 33-45.

REP: To address this comment, we have added a phrase: “In recent years, some progress in earthquake recording along the Gakkel Ridge was achieved owing to the installation of several seismic stations on islands in the Arctic Ocean (e.g., Antonovskaya et al., 2020)” (L46-48).

Lines 49-50: is it really a frequent feature? I would tone down on the “frequent” and remove it. Some studies show otherwise, i.e., < 18 km depth, e.g., Grevemeyer, I., et al., 2019.

REP: We have corrected this sentence as “Similarly deep seismicity is observed in a few other centres of slow spreading and is explained by anomalously low temperature beneath the rift axis (13,14,15)” (L51-53).

Line 57: change “contaminated” by “enriched”

REP: Corrected

Line 60: Why is the authors stating that seismicity reaches 25 km depth when they show on figure 2 (red dots) that some EQs were recorded at 32 km focal depth (which is impressively deep and not discussed)?

REP: Here we write the phrase: “**Robustly resolved seismicity** was detected at depths of up to 25 km, which are far below the bottom of the crust (21)” (L63-64). This is based on the work of (21), where there is seismicity down to 25 km depth, with a with a few scattered events deeper than that. These results were based on a 1D velocity model initial location. We now consider a 3D velocity model and obtain a few events deeper than 25 km. In the later parts of the article, we will admit that these events appearing at greater depths are less robust, and should be interpreted with prudence (L139).

Lines 114-120: this seismic gap inferred and interpreted as the consequence of crustal rocks being too soft (surface: more fractured and saturated with seawater and in depth: remnant magma pocket), which would prevent accumulation of stresses and generation of seismicity, have been observed in Schlindwein and Schmid’s (2016) study of SWIR micro-seismicity. Later,

Grevmeyer et al. (2019) analysis of the SWIR dataset indicates that the previously proposed thick aseismic region in the upper lithosphere was “simply a consequence of a model embedded in the hypocenter location procedure that did not include a few 100 m-thickness of unconsolidated sediment”. They show the strong influence of such layer (and hence the importance of including such layer in models) on the focal depth and apparent seismic gap. Furthermore, because of their very slow spreading rate, ultraslow-ridges are known to have substantial sediment thick layer at the axis and flanks.

Was any unconsolidated sediment layer considered in this present study to avoid this bias which would result in having artifacts, i.e., seismic gap and deep seismicity? If not, how can you state that this isn't a bias of not considering such layer and can you provide model results including such layer for comparison?

REP: To address this comment, we have added a series of new synthetic tests presented in Figure S8, in which we evaluate a possible role of the upper low-velocity layer on the results of source locations. The results of this test demonstrate that the completely different structures of the upper layer in the rift valley do not dramatically affect the distributions of events. In particular, in the model, there were several shallow events that remained at approximately same depths after recovery, regardless the uppermost structures. Thus, the suggestion about critical effect of sediments on the rift valley on source locations appears to be not valid. We have paragraph with the discussion of this issue in Supplementary in L239-249.

Note also that the layer of soft unconsolidated silica ooze is very special to the site at the SWIR (located beneath the Polar Front and acting as sediment trap) and there it only affected stations in the rift valley. These soft sediments were visible in parasound data and even in seafloor imagery. In the seismicity data, significantly delayed S phases appeared on the stations affected by the soft sediments, but not on the remaining stations outside the central rift valley. At Gakkel Ridge, such a layer of silica ooze has not been observed. Instead there is a thick pile of comparatively consolidated sediments of continental origin covering the rift valley away from the volcanoes. None of the recently refraction seismic studies (Jasmine cruise, AMORE) gave indications for such a layer. Furthermore, we currently process OBS data from a volcanic center at 120°E Gakkel Ridge. No indications for anomalous delays of S phases are visible there. Along Knipovich Ridge, for example, there are no unconsolidated sediments at the surface and we used a similar location routine (comparing different approaches) for an extensive along axis network of seismometers. We observe a) an aseismic upper lithosphere with temperature-controlled boundaries and an onset of seismicity below 10 km depth – comparable to the observation at the SWIR, which by the way persists after dealing carefully with the soft sediment layer – see comment to Grevmeyers paper. b) seismicity generally is not recorded from the top 3-4 km below the surface, probably due to the processes discussed here – fracturing, presence of sea water etc. In this paper it becomes obvious that in some places there is scattered very shallow seismicity.

We are therefore certain that we see here an aseismic behaviour that is different from the one described in Schlindwein and Schmid and Meier et al. that is related most likely to aseismic deformation of altered mantle material in amagmatic regions. Instead we rather observe a general lack of very shallow seismicity (as in Meier et al.) with some scattered shallow events, that is mostly a result of local surface conditions and potentially the detection threshold of the network for weak, shallow seismicity.

Line 125: here again, a depth of about 25 km is mentioned while they clearly show deeper seismicity (up to 32 km) – is the 32 km EQ(s) considered an outlier(s)? If so, it needs to be explicated and discussed.

REP: To address this comment, we have added a phrase “There are a few deeper events down to ~32 km, but their locations are less robust and should be considered with prudence” (L136-137).

Line 127: this is a highly speculative statement. I would rephrase it: “We speculate a fault crossing the entire lithosphere”.

REP: We have rephrased this statement as: ” There are a few deeper events down to ~32 km, but their locations are less robust and should be considered with prudence” (L138-139).

While Tao et al., 2020 paper focuses more on the coupling between seismicity and hydrothermalism, some of their results aligned with this paper findings should be acknowledged here, i.e., showing a deep detachment fault reaching 15 km deep associated to deep seismicity, and asymmetry.

Tao, Chunhui, W. E. Seyfried, R. P. Lowell, Yunlong Liu, Jin Liang, Zhikui Guo, Kang Ding et al. "Deep high-temperature hydrothermal circulation in a detachment faulting system on the ultra-slow spreading ridge." Nature communications 11, no. 1 (2020): 1-9.

REP: Thanks a lot for this comment. We found this reference absolutely relevant to our case. We have added a sentence mentioning this work (L141-145).

Line 137: what does “self-consistent” means here?

REP: We have removed these words without loss of the meaning.

Line 137: is it not thermo-mechanical numerical modelling to explore magmatic processes rather than a magmatic-thermo-mechanical numerical modelling? See your line 178.

REP: The word “magmatic” has been removed from this sentence.

Lines 150-151: some kind of pleonasm here - by definition if a section is amagmatic then the magmatic activity is rare. Perhaps reformulate this sentence to avoid the redundancy.

REP: We have removed the word “amagmatic”

Lines 151-152: is it the spreading ridgeS as in several ridgeS succeeding each other or is it one ridge (Gakkel) with several segments/sections that are not dissected by transform fault? It needs clarification.

REP: We have rewritten this sentence as: “Our numerical experiments thus suggest that the Gakkel spreading ridge is developing under the combined effects of an ultraslow spreading rate and low mantle potential temperature and consists of several successive segments, which are connected by wide, oblique amagmatic sections” (L180-183).

Line 183: replace “which corresponds” with “which would correspond”

REP: Corrected

Line 187: remove “outstanding”. Deep seismicity and asymmetry are common to all ultra-slow spreading ridges (including the Mohn and Knipovitch ridges) as stated by the authors in the introduction who referenced all previous work done at these ridges.

REP: We do not agree. The activity that occurred in 1999 on the 85E segment was really outstanding regarding the intensity of volcanic processes and magnitudes of earthquakes.

Lines 196-197: this is inaccurate. Define “regular spreading zones”. The statement saying that “magma generation occurs continuously along ridges” is only true for fast-spreading ridges which are not accounting for all the “regular spreading zones”. Both slow-spreading and ultra-slow spreading ridges have discrete magmatic zones along segments/sections. It is not a specific characteristic of the Gakkel per se but rather a shared characteristic with the others ultra-slow and slow-spreading ridges to some extent. Please refer to Cannat’s 1993 paper and all the

literature that followed. Please rephrase your statement.

Cannat, M. (1993). Emplacement of mantle rocks in the seafloor at mid- ocean ridges. *Journal of Geophysical Research: Solid Earth*, 98(B3), 4163-4172.

REP: We have reformulated this phrase as: “Because of such a low degree of melting, the magma beneath the Gakkel Ridge is focused in discrete zones that are separated by long amagmatic segments. This appears to be a common feature for most ultraslow and slow spreading centres, in contrast to fast-spreading zones, where magma generation occurs continuously along the ridges (33).” (L234-237)

In Supporting Information:

Lines 9-11: This statement is not quite accurate. Spreading rate for Mohn and Knipovitch ridges is about 15-17 mm.a-1 (Bruvoll et al., 2009), which is < 20mm.a-1, hence these two ridges belong as well to the ultraslow spreading ridges. Please reword.

REP: This depends a little bit, what one calls ultraslow. Some people require ultraslow spreading ridges to open at less than 12 mm/y. Also spreading rates differ a little bit according to the model used. Knipovich Ridge because of its obliquity and hence very low effective spreading rates certainly shows ultraslow characteristics. According to this comment, we have modified this sentence as: “Such slow values of the spreading velocities are compatible with only a few locations in the world, such as Mohn and Knipovich Ridges (15-17 mm/year) (40) and Southwest Indian Ridge (~14 mm/year) (10,41).” (L9-11 of Supplementary).

Lines 17-19: here again this statement is not quite accurate. Gakkel is not the only one not being dissected by transform faults - both Mohn and Knipovitch ridges share this “difference”. Please reword.

REP: We have reformulated this phrase as: “Another feature of the Gakkel Ridge is that its axis line is smooth and almost not dissected by transform faults (10). Such a behavior is only observed on a few other ultra-slow ridges (such as Mohn and Knipovich Ridges), but it appears to be exceptional compared to other spreading centres.” (L17-20 of supplementary)

Reviewer #3 (Remarks to the Author):

The manuscript tackles a major unsolved issue of mid-ocean ridge research, i.e., the relationship between low-degree of mantle melting and the occurrence of explosive volcanism along the ultraslow spreading Gakkel Ridge in the Arctic Ocean. The authors provide constraints from seismic tomography – revealing structural elements and numerical modelling – indication/simulating potential processes. In concert, both may provide indeed a new perspective on the mechanisms controlling the rather puzzling behaviour of ultraslow spreading ridges, which – for a long time – were considered to be rather amagmatic and therefore scientists were rather surprised to find explosive volcanism and patches of robust magmatic accretion. In general, I therefore believe that the manuscript can make an important contribution.

However, there are a number of features that need to be improved before the manuscript can be finally evaluated. First, the text and title are still rather “sloppy”. For example, the title highlights the features to be discussed in the paper, but overall, it’s nothing new.

REP: In our opinion, the statement in the title, which is proven by our results, is not obvious. As far as we aware, the link between “low-degree mantle melting” and “explosive volcanism was not explicitly considered by anybody before.

Thus, rocks showing a low degree of mantle melting were sampled years ago. And of course – the mantle melting is the underlying process that controls features occurring at shallower level – like deep seismicity and occurrence of explosive volcanism. The fact that seismicity shows a very deep level of activity was shown by one of the authors previously (Korger and Schlindwein, *Geophys. J. Int.*, 2012) and also the occurrence of explosive volcanism is well known (Sohn et al., *Nature*, 2003).

REP: Yes, the information you mentioned was known before. We have just added a few new elements in this puzzle, namely P-S tomography and numerical modeling, that together with the previously known data has allowed us to create a composite picture and to propose some not obvious conclusions.

When reading the manuscript, I'm lost if we really learn something new. Therefore, the authors fail to make their point clearly. However, some of my personal uncertainty about the manuscript comes from the style of writing and how terms are chosen/used. Thus, at several places non-unique geophysical observations from the tomography are presented as supporting a unique explanation.

For example, high Vp/Vs ratios are presented of being a solid fact for melting. Of course – at settings where melts migrate upwards geophysical data often reveal high Vp/Vs ratios. However, also settings with a high degree of serpentinization show high Vp/Vs ratios (Christensen, 2004) and serpentinization is a common process at ultraslow spreading ridges. Further, a fractured rock with fluid filled veins also has a high Vp/Vs ratio (Popp & Kern, 1994; Wang et al., 2012). Thus, it's rather misleading to claim that Vp/Vs provides unique evidence for occurrence of melting – at least in the lithosphere of an ultraslow spreading mid-ocean ridge. **REP:** This is not fair. Besides melting, our interpretation also presumes the existence of volatiles, as stated throughout the text. To set this statement clearer, we replaced “*partially molten volatile-rich magma*” with “*partially molten and/or volatile rich rocks*”, L20. We mention both melts and volatiles in L105, L119 and others). However, the numerical simulations may suggest that at the depth/location where the tomography finds high Vp/Vs ratios, melts should occur. In this case, the interfingering of simulations and evidence from the seismic tomography would support a new point of view that may indeed merit publication. Unfortunately, the current manuscript isn't making the point – or if it does, it's not well presented. Thus, to really make the paper a manuscript of interest to a broad audience, results from the seismic tomography should be linked in the discussion much more clearly to the features of the numerical model. Further, observation should be presented in a way that alternative interpretations are not ignored (like high Vp/Vs could be either melts, serpentinized rock or fractured and fluid filled rocks – or a combination of all).

REP: We agree that besides melt content, many other factors can affect the seismic velocities and Vp/Vs. Obviously, it is not possible provide a unique interpretation based on only seismic models. Throughout the text, we softened the statements related to interpretation of our model by saying that this is one of possible explanation of the observed structures.

At the same time, we slightly disagree with a possibility to interpret the high Vp/Vs ratio by serpenization instead of melting. Of course, at ultraslow spreading ridges, serpentinization plays a role, but away from such large volcanic centers with fresh basalt at the seafloor and a thick magmatic crust. With the Jasmine cruise and findings at the SWIR, there is growing evidence that the volcanic centres of ultraslow spreading ridges do show a crust that is much thicker than in the adjacent amagmatic regions. In addition, at the Seg8 volcanic complex, for example we found this melt volume based on seismic tomography and high vp/vs ratios and we see a very comparable feature at Logachev volcano on Knipovich Ridge (recently submitted seismic tomography). We don't think that anyone would argue for serpentinization in these regions that show concurrent intrusive activity with migrating earthquake swarms. So we think we need to stress this

point more that we see high V_p/V_s ratios immediately below a region that is known to show recent (1999-2001) volcanic activity.

Further, the relationship between features found in the seismic tomography and the numerical simulations need to be explored in more detail. In the current version of the manuscript I have sometimes the feeling that the comparison of features is rather incidental (i.e., comparing only one or two depth slices instead of using the geometry of a 3D volume). Thus, to relate the pattern found in the simulations to features found in the tomography, you only used one depth slice at 20 km. What will be needed is to show it in 3D.

REP: Unfortunately, it is not possible yet to set the direct connection between the tomography results and numerical modeling. It is clear that seismic tomography cannot provide unique values of physical parameters needed for modeling due to many reasons (lack of resolution, uncertainty in damping definition, non-unique transition from seismic to petrophysical properties etc). On the other hand, numerical modeling presumes some simplifications that makes impossible to simulate directly all details observed in the nature. Finally, the numerical modeling provides temporal evolution, whereas seismic tomography is an instantaneous snapshot of the current state. Therefore, we can compare seismic tomography and numerical modeling results only qualitatively. In our opinion, demonstration of the correspondence in 2D sections is clearer than analyzing 3D images. Presenting tomography results in 3D is only efficient if there is a possibility of interactive rotation of the model; just static images of the 3D patterns often look non-informative and sometimes misleading.

In summary, I do see a paper with interesting and new results from the Gakkel Ridge, providing great structural images from seismic tomography. Further, numerical simulations are used to yield (some of) the processes controlling the behaviour of mantle melting and thus both datasets together can provide a new and deeper understanding. However, the current paper fails to do it. First, in the introduction the open questions and mysteries of the Gakkel Ridge should be clearly stated, to help readers to understand the aim of the study and why the different approach were used and why they nurture each other. Right now, the aim of the study isn't really stated.

REP: We have made corrections throughout the text of the main paper to better emphasize the importance of joint using seismic tomography and numerical modeling.

Second, results from the tomography should be discussed more openly. Do not try to “hide” that some interpretations might not be unique.

REP: In the supplementary, we have added a lot of information with the details on how seismic tomography was performed that honestly show the limitations and weak points of our seismic model

Third, describe more clearly the features arising from the simulations. To me it's unclear if melt migration is really a part of the output of the modelling or if it is approximated from the other parameters (note - I'm not a modeller myself and thus it should be clear to “dummies”).

REP: The modelling results have been described more clearly. Our model simulates melt extraction and propagation in a simplified manner (see Method section). The computed melt trajectories follow the topography of the mantle solidus surface under the lithosphere, which allow melt accumulation in multiple magma chambers that develop in the areas of the highest solidus surface topography (magmatic segments). Numerical results are shown in the Fig. 3. The lowered melt supply associates with strong variations of brittle-ductile boundary depth along the ridge, which therefore breaks spontaneously into narrower and hotter magmatic and wider and colder amagmatic sections. The oval-

shaped magma chamber spontaneously forms by focusing of the extracted low-degree melts toward individual melt traps. Melt migrates along the detachment faults in the footwall and normal faults in the hanging wall and then forms new crust and volcanoes through spontaneous cooling and crystallization

Last, discuss both approaches in concert and thus show readers why your results are so cool to merit publication in Nature Communication. What's the ground-breaking result? As I iterated above – all features from the title have been known before. So – why your study and what's the benefit?

Additional comments (mostly taken from the annotated m/s; please note the m/s has more comments and suggestions and I only copied a few into this formal review):

1.) Some of the SP waves in Supplementary Figure 1 show SP waves which have larger amplitudes than P - as expected. However, some have smaller amplitudes. Are those really SP waves or are they some P-wave multiples of sidescapes and not converted S-wave energy? Please provide Wadati plots (or at least one plot combining P-time vs. S-time of all pairs) in the supplement. A Wadati plot of each earthquake should help to define better if picks are indeed appropriate or spurious.

REP: Wadati diagrams are in this case difficult to use. The concept of the Wadati diagram presumes a model with a constant V_p/V_s ratio, which is obviously not our case. In our model, $V_p/V_s=1$ in water and is in a range of 1.7-1.9 in the solid ground. Therefore, for a shallow local event, the Wadati diagram will be nearly horizontal. The angle of the diagram would be mostly dependent on the depth and remoteness of the event, and not on the actual V_p/V_s ratio. It would give very strong scattering of the Wadati diagram making it useless.

Furthermore, since the stations drift on sea ice over an area with highly variable bathymetry, a different thickness of the water layer needs to be incorporated into the Wadati plots, requiring some knowledge of the wave propagation paths already. In addition, in quite a few cases, we receive clear P arrivals from all stations but only SP arrivals on one of the 3 arrays, such that a Wadati diagram (requiring at least 2 SP-P/P pairs) cannot be established.

According to this reviewer's comment, we have considerably expanded the description of the initial data processing in L107-125 of Supplementary.

2.) Please also show histograms of residuals of both P- and SP. The quality of tomographic results is rather poorly presented.

REP: In supplementary, we have added a table S1 with the information about the mean residuals during iterations and some text with the discussion of the observed values (L169-178). According to the reviewer's comment, we have added Figure S2 with histograms of the residuals after the 1st and 4th iterations.

3.) Korger and Schlindwein (GJI, 2014) provided a P-wave tomography from the very same data used in this study. I have to say that the transects from both studies are rather difficult to compare as the strike of depth sections is slightly different. However, Korger and Schlindwein didn't reveal any P-wave anomaly where this study has a strong positive velocity anomaly. Which tomography shall I trust? Rather puzzling and I would like to get a good explanation for the different features stemming from the same data. Considering that travel time data are the same I cannot see why the models should be so much different.

REP: The distribution of the P-wave velocity anomalies in the new tomography model demonstrates some similar features compared to the previous model by Korger and

Schlundwein (2014). At the same time, some changes in the velocity model are caused by considerable improvement in source location accuracy owing to adding the S-wave data.

Korger and Schlundwein (2014) did a first relocation of hypocentres in a 3D velocity model that only incorporated a 3 D bathymetry layer. After that, since FMTOMO can only invert for P velocity, earthquake locations were kept fixed and the P velocity model was determined. In view of much better determined P phases, this seemed an appropriate choice. LOTOS has the ability to invert jointly for S and P velocity structure and apparently can also deal well with data sets that only have limited observations. So this new study is certainly a more complete image of the local structure and given the pronounced difference in v_p/v_s ratio, a joint inversion and certainly a relocation of the events in a new 3D P and S velocity model is necessary. We have added a few phrases about the comparison of these studies in Supplementary in L162-168.

4.) Why occur some V_p/V_s anomalies with a similar amplitude away from the proposed magmatic conduit (to the north and west)? If it is really magmatic activity causing the anomaly and the activity is linked to the local earthquakes - why have the other anomalies no quakes? Or are these features not so well resolved and hence artefacts? Based on Korger and Schlundwein only the central area had a large number of rays and good coverage. How much bias is caused by rays having longer offsets. For those, the tomography will have a rather low resolution at shallow depth. However, shallow features are important and show generally the largest lateral variations, causing significant delays. How robust is the inversion of deep-seated features when the shallow structure lacks resolution/coverage?

REP: We have added some new synthetic tests with free-shaped anomalies to see the robustness of the anomalies in different parts of the study area. It can be seen that the high-amplitude anomalies in the western margin of the area are not correctly resolved. Therefore, according to the results of these tests, we masked the poorly resolved areas in the main results. Regarding the deep-seated anomalies, we have designed a special test in Figure S7 to assess their stability. We have added the descriptions of these tests in L226-249.

5.) Please discuss what is the origin of the other V_p/V_s anomalies? Melts, too? Fracturing?

REP: We admit that the numerical values of anomalies strongly depend on many different factors, such as uneven resolution and uncertainty of damping definition. Therefore any attempts of direct conversion of seismic parameters to temperature, melting, composition and other petrophysical parameters are very risky. Therefore, we insist that the results of seismic tomography in most cases can be interpreted only qualitatively. It should be noted that our resulting distribution of V_p/V_s looks similar to the results of tomography inversions for other active volcanoes in the world, which allows us to make some extrapolations. We have added a paragraph with the discussion of this issue in the main article in L105-113.

6.) The swath mapping had a rather limited coverage – only at the centre of the study area. How was the seafloor depth constrained where no swath data exists? From IBCAO2 bathymetry? In this case ray entry points into the seafloor might be biased as the seafloor is often based on interpolated data and not measured soundings, in turn, causing significant delays and hence limiting the resolution power of the tomography.

REP: For calculation of travel times, we used lower resolution regional model presented in Figure 1c. It is true that using this oversmoothed model may cause some bias of travel times. On the other hand, some natural averaging occurs due to finite frequency content of the arriving waves. For example, for periods of 0.2-0.5 s and velocity 5-7 km/s, the wavelength would be in the range of 1-2 km. Therefore, the conversion of seismic waves

on the sea bottom will occur not in a point, but in an area of ~1 km size. For this reason, the converted travel times would not be strongly affected by high-resolution patterns presented in Figure 1c. We have added a few phases on this issue in Supplementary in L135-141.

7.) In the supplement it is described that the water layer is approximated by a constant velocity. However, the water in the arctic is strongly layered (water mass with very little salinity at the top, Atlantic bottom water at greater depth which a much higher salinity and possible some mixing in-between). I guess that layering of the water layer will have some effect on the location of ray entry points into the seafloor. Considering the large variations of the seafloor this should cause some bias. Why not considering the real layering of seawater in the tomographic model?

REP: Yes, it is true, in the ocean water, there are some velocity changes of a few percent in a range of 1490-1520 m/s (Frosch, 1964) that may affect travel times of seismic rays. The major changes occur in the vertical direction due to depth stratification of salinity and temperature. If we knew the actual 1D velocity profile in the water and could use it to calculate more accurately the rays, the travel times would be biased in one direction to approximately same value. Regarding the uncertainty of the origin time determinations of seismic events, this nearly constant bias would not affect the results. The role of lateral heterogeneities cannot be estimated, because their distribution is impossible to identify. It should be noted, however, that even in much more sophisticated seismic surveys conducted offshore for hydrocarbon exploration, water heterogeneities are considered as secondary effects and are not taken into account in standard processing. We have added some discussion on this issue in supplementary in L142-149.

Frosch, R. A. (1964). Underwater Sound: Deep-Ocean Propagation: Variations of temperature and pressure have great influence on the propagation of sound in the ocean. *Science*, 146(3646), 889-894.

8.) Seismicity in the sections is plotted without error bars and interpreted as being very precise (discussed in terms of indicating a fault zone). However, even in a perfect Earth travel times are associated with some uncertainty when picking onsets in noisy data. First, please report the magnitude of errors for P- and S-wave onsets (at least in the supplement). Further, how large is the horizontal and vertical uncertainty? Please plot error bars from the location procedure. At least indicate the average uncertainty (as is often done in similar studies).

REP: To address this comment, we have added Figure S9 with the distribution of event mislocations in synthetic tests, which adequately represent the uncertainty of source locations. This figure includes the location results in the starting 1D model and in the final 3D model with indications of average location errors. In our opinion, this representation is more adequate than showing error bars provided by standard location tools. We have added some text to describe this result in Supplementary in L250-257.

9.) Line 139-141: “The modeling results show that the ultraslow spreading rate and low mantle potential temperature are critical to the formation of a very low magma supply by volatile-rich low-degree melts (7,8) derived from the slowly rising and decompressing asthenosphere.” – awkward sentence. Either this are your results (in this case you do not need a reference to any paper) or it’s from the cited papers. In this case it won’t be your results. This is a good example of what I considered “sloppy language”, which leaves reader wondering about the results. If your study and the other paper(s) show the same – just state it like that.

REP: The sentence has been clarified.

10.) The numerical modelling results are difficult to place. Here, only results from ultraslow spreading rates are presented. If the findings are so unique to ultraslow spreading ridges, your

case would be much stronger when you show that it's not occurring at slightly faster spreading rates and it would also be nice to learn at which spreading rate the transition from one mode to another occurs.

REP: Models with a wide range of spreading rates and mantle potential temperature were conducted. With spreading rates increasing (see the figure below), even though the magma supply is very low due to the low mantle potential temperature, the amagmatic section still disappears at the intermediate spreading rate (40 mm/yr). Furthermore, with the mantle potential temperature increasing, even though the spreading rate is ultraslow (10 mm/yr), owing to the high magma supply, the amagmatic section also still disappears at the high mantle potential temperature. Thus, the Gakkal ridge occurs under the combined effects between ultraslow spreading rate and low mantle potential temperature.

Models with different spreading rates. Low mantle potential temperature (1255 °C) is implemented in these models.

Models with different mantle potential temperature. Ultraslow spreading rate (10 mm/yr) is implemented in these models.

11.) I added a comment to the m/s that the process that leads to the accumulation of volatiles needs to be explored in much more details. I believe that this is really one of the most important features, highlighting the link between ultraslow seafloor spreading and explosive magmatism. Right now, it's just one of many features in the m/s and not well exposed.

REP: Thanks a lot for your comments. We agree that this feature is important and it is highlighted throughout the paper, including the abstract and the conclusion paragraph.

12.) The numerical model reveals very thick crust – up to 20 km thick. This is not consistent with observations showing very thin crust in the Arctic (Jokat & Schmid-Aursch, GJI, 2007) and the conceptual model presented in Fig. 4. Please explain. Or is the numerical model using a different definition of crust? However, this is rather awkward.

REP: Short discussion of crustal over-thickening is added. The large crustal thickness in magmatic segments is likely due to over-focussing of crustal growth by the current simplified melt extraction and transport algorithm (see Methods), which brings all extracted melts to very localised magma chambers forming at the maxima of mantle solidus surface topography. No melt is allowed to be emplaced into the crust outside of these chambers. In the future, melt transport and emplacement algorithm need to be improved (e.g. by implementing more distributed melt emplacement via dikes).

I added in total 69 comments (some just minor grammatical features) to the annotated m/s. Please consider those when revising the manuscript. I believe that the content of the m/s is of great interest and will have a considerable impact – when presented differently. This is perhaps just my personal point of view and I might be mistaken, driven by my ignorance. However, when I have difficulties in following the m/s – I'm afraid that others will have similar difficulties, too.

REP: Thanks a lot for your careful consideration, which was very useful to improve our paper.

Good luck with the revision, Ingo Grevemeyer

Additional references not in the manuscript but cited in my review:

Christensen, N. I. (2004), Serpentinities, peridotites, and seismology, *Int. Geol. Rev.*, 46, 795–816, doi:10.2747/0020-6814.46.9.795.

Popp, T. and H. Kern (1994), The influence of dry and water saturated cracks on seismic velocities of crustal rocks – A comparison of experimental data with theoretical models, *Surv. Geophys.*, 15, 443-465.

Wang, X.-Q., A. Schubnel, J. Fortin, E. C. David, Y. Guéguen, and H.-K. Ge (2012), High Vp/Vs ratio: Saturated cracks or anisotropy effects?, *Geophys. Res. Lett.*, 39, L11307, doi:10.1029/2012GL051742.

REVIEWERS' COMMENTS

Reviewer #1 (Remarks to the Author):

The authors were fortunate to get three thorough reviews, and they have taken advantage of them to revise the paper and greatly improve it. I think the paper can now be published in its present form.